# Spermatozoal Mitochondrial Dynamics Markers and Other Functionality-Related Signaling Molecules Exert Circadian-like Response to Repeated Stress of Whole Organism

**DOI:** 10.3390/cells11060993

**Published:** 2022-03-15

**Authors:** Isidora M. Starovlah, Sava M. Radovic Pletikosic, Tamara M. Tomanic, Marija L. J. Medar, Tatjana S. Kostic, Silvana A. Andric

**Affiliations:** Laboratory for Reproductive Endocrinology and Signaling, Laboratory for Chronobiology and Aging, CeRES, DBE, Faculty of Sciences, University of Novi Sad, 21000 Novi Sad, Serbia; isidora.starovlah@dbe.uns.ac.rs (I.M.S.); sava.radovic@dbe.uns.ac.rs (S.M.R.P.); tamara.tomanic@dbe.uns.ac.rs (T.M.T.); marija.medar@dbe.uns.ac.rs (M.L.J.M.); tatjana.kostic@dbe.uns.ac.rs (T.S.K.)

**Keywords:** repeated psychological stress response, mitochondrial dynamics and functionality markers, cAMP signaling markers, MAPK signaling markers, circadian, spermatozoa number and functionality

## Abstract

In the search for the possible role of the mitochondrial dynamics markers in spermatozoa adaptation, an in vivo approach was designed to mimic situations in which human populations are exposed to 3 h of repeated psychological stress (the most common stress in human society) at different time points during the day (24 h). The hormones (stress hormone corticosterone and testosterone), the number and the functionality of spermatozoa (response to acrosome-reaction-inducer progesterone), as well as the transcriptional profiles of 22 mitochondrial dynamics and function markers and 22 signaling molecules regulating both mitochondrial dynamics and spermatozoa number and functionality were followed at three time points (ZT3, ZT11, and ZT23). The results show that repeated stress significantly decreased the number and functionality of spermatozoa at all time points. In the same samples, the transcriptional profiles of 91% (20/22) of mitochondrial dynamics and functionality markers and 86% (19/22) of signaling molecules were disturbed after repeated stress. It is important to point out that similar molecular changes in transcriptional profiles were observed at ZT3 and ZT23, but the opposite was observed at ZT11, suggesting the circadian nature of the adaptive response. The results of PCA analysis show the significant separation of repeated stress effects during the inactive/light and active/dark phases of the day, suggesting the circadian timing of molecular adaptations.

## 1. Introduction

Mitochondria are complex, highly dynamic, intracellular organelles that have a central role in cell physiology. Mitochondria do not have a stable shape (they stretch, shrink, blend, and divide all the time), and energy production is not their only role. They are involved in a variety of biological functions, predominately supporting critical energy power needs by oxidative phosphorylation of the electron transport chain, but also steroid and stress hormone production, ion homeostasis, apoptosis, reactive oxygen signaling, etc. [1,2]. Like those of somatic cells, mitochondria in spermatozoa are essential for cell life too. These organelles form tight helices at the mid-piece of sperm during spermatogenesis and contribute to the functionality and motility of sperm—a highly energy-driven and demanding process [3]. Mitochondria contain their own genome (mitochondrial DNA, mtDNA) which encodes a limited number of proteins. Thus, various mitochondrial disorders and mtDNA mutations in somatic cells are found to be associated with a wide spectrum of diseases. Additionally, mitochondrial disorders in spermatozoa are strongly related to male infertility [4,5].

Recent research in spermatozoa physiology is focusing on the powerhouse of the cell, a mitochondrion, as a potential biomarker of sperm health and fertility. The functionality of mitochondria determines human spermatozoa with high and low fertilizing capability [6]. Moreover, the stages of spermatogenesis are characterized by changes in mitochondrial morphology [7]. Mitochondrial functionality might be necessary to maintain sperm acrosin activity, acrosome reaction, and chromatin integrity [8]. There is also a connection between defects of mtDNA in oligoasthenozoospermic patients and poor diagnostics [9]. mtDNA depletion plays an important role in the pathophysiology of male infertility [10] as a diagnostic marker of sperm quality in infertile men [11]. Large-scale deletions of mtDNA were indicated as risk factors for poor sperm quality in asthenoteratozoospermia-induced male infertility [12]. Sperm mtDNA copy number has also been utilized as a biomarker of male reproductive health and the probability of pregnancy success in the general population [3]. According to all of the research mentioned above, the mitochondrial network homeostasis is essential for male fertility. It is kept and maintained by a well-coordinated process of mitochondrial dynamics, including complex mitochondrial protein-import machinery (mitochondrial transduceom), the movement of mitochondria to position themselves strategically in the cell (motility/trafficking), mitochondrial biogenesis, mitofusion, mitofission, and mitophagy [1,13,14,15]. More importantly, all signaling pathways regulating mitochondrial dynamics are essentially involved in the regulation of spermatozoa function. A great gap and interest in research related to mitochondria and male fertility have recently been indicated [16].

Stressful life has been recognized as one of the main reasons for male infertility [17,18,19], which is a global problem showing an increase in unexplained cases of infertile young males. Different types of stressors and stressful life events have been linked to reduced male reproductive function [18,20,21,22]. A high number of stressful life events are observed in infertile men, and this was associated with a decline in semen quality during fertility treatment [18]. Mitochondria are the key linking point between stress response and spermatozoa functionality since they are responsible for satisfying enormous energy demands required for both processes [17,23,24,25]. Stress enhances rat testicular germ cell apoptosis [26] and the irreversible loss of germ cells and spermatozoa number [27]. Furthermore, both stress signaling and mitochondria are essential for spermatozoa functionality. Epidemiological studies showed that DNA damage during stress response is regulated through β2-adrenergic-receptors [28]. Moreover, fertility and spermatogenesis are altered in α1-ADRs-knockout-male-mice [29], suggesting the importance of stress signaling in the regulation of mitochondrial homeostasis. 

Among the many stressful challenges which modern human society is facing are night work or work in shifts, “jet-lag”, too much time spent indoors, etc. This departure from the way of life we are evolutionarily adapted to can lead to many health disorders but also affect reproductive ability [30,31]. The photoperiod dependency of sperm DNA synthesis and spermatogenesis has been known for many years. Previous epidemiological data, as well as genetic studies in humans and animals, support the contribution of the circadian clock in male fertility. The circadian clock, which is a temporal program that developed as an adaptation to Earth’s rotation, is one of the most basic physiological mechanisms exerting its effect through the control of metabolism, endocrine and immune function, as well as behavior [32]. The oscillation of different factors that coordinate the spermatogenic wave differs from the circadian transcriptional–translational feedback loop that is not cell-autonomous but requires the integration of many steps that occur in different cell types (reviewed in [33]). The stable temporal pattern of spermatogenesis is probably a consequence of the highly constant duration of germ cell proliferation and differentiation, but it is not yet completely understood [34]. Almost three decades ago, a diurnal rhythm of murine spermatogenic DNA synthesis was reported in NMRI mice: the highest proportion of DNA-synthesizing cells (mainly spermatogonia and preleptotene spermatocytes) was seen at 8 p.m. and especially at 10 p.m., while the lowest proportion was observed at 2 p.m. [35], supporting the circadian pattern of spermatogenic waves.

Male fertility and semen quality are not only important markers of reproductive health but also are the fundamental biomarkers of overall health [17]. There is a critical need for the accurate assessment of male fertility for estimating overall reproductive health considering serious limitations of conventional semen analysis [3]. Although many studies have suggested the correlation between a stressful life and male (in)fertility [1,13], the mechanisms are not described yet. Here, we hypothesize that psychophysical stress causes different changes in mitochondrial dynamics and functionality markers, as well as related signaling molecules, depending on the circadian time of the stress exposure. The effects of the repeated stress were followed at different time points during the day (light/inactive and dark/active phase): ZT3—3 h of stress started at 7 a.m. (ZT0, lights on) and finished at 10 a.m.; ZT11—3 h of stress started at 3 p.m. (ZT8) and finished at 6 p.m.; ZT23—3 h of stress started the next day at 3 a.m. (ZT20) and finished at 6 p.m. (please see Figure 1; ZT—zeitgeber (time giver)). Our intention was not to study spermatogenesis but only the patterns of the transcriptional profile of mitochondrial dynamics markers as well as markers of signaling pathways (cAMP and MAPK signaling) related to the regulation of both mitochondrial dynamics and spermatozoal functionality. We believe that this approach will reveal the pattern of molecular adaptation of spermatozoa and help in the development of new molecular markers for the assessment of male (in/sub)fertility.

## 2. Materials and Methods

All samples, commercial reagents/assays, primers, and software that were used in this study are given in the Appendix A.

All experiments were performed in the Laboratory for Reproductive Endocrinology and Signaling and Laboratory for Chronobiology and Aging, Faculty of Sciences at University of Novi Sad (https://wwwold.dbe.pmf.uns.ac.rs/en/nauka-eng/lares, accessed on 4 March 2022). All the methods used in this study were previously reported by our group (for all references, please see [21,22,36]) and followed the relevant guidelines and regulations.

### 2.1. Statement of Institutional Review Board

The Committee of the Faculty of Sciences, University of Novi Sad, Novi Sad, Serbia, approved the manuscript.

### 2.2. A Statement That the Authors Complied with ARRIVE Guidelines and Institutional Animal Care and Use Committee Guidelines

The authors complied with ARRIVE guidelines, and all experiments were in adherence to the ARRIVE guidelines. Furthermore, all experimental protocols were approved (statement no. 01-201/3) by the local Ethical Committee on Animal Care and Use of the University of Novi Sad operating under the rules of the National Council for Animal Welfare and the National Law for Animal Welfare (copyright March 2009), following the NRC publication Guide for the Care and Use of Laboratory Animals and NIH Guide for the Care and Use of Laboratory Animals. 

### 2.3. Animals and Experimental Model of Stress

All experiments were carried out using adult, three-month-old male *Wistar* rats. Animals were bred and raised in the accredited Animal Facility of the Faculty of Sciences, University of Novi Sad, Serbia, in controlled environmental conditions (22 ± 2 °C; 14 h light and 10 h dark cycle, lights on at 7:00 a.m.) with food and water ad libitum. The experimental model of psychophysical stress by immobilization was performed using the method previously described [20,21,22]. In short, stressed (3hIMO) rats were bound in a supine position to a wooden board by fixing the rats’ limbs using thread, while the head motion was not limited. Freely moving, unstressed rats were used as a control group in each experiment. All the activities during the dark phase were performed under red light. To analyze the effects of the repeated immobilization stress at different times during the day (Figure 1), animals were subjected to immobilization stress for 3 h for 10 consecutive days (10×3hIMO) in different periods during 24 h (from ZT0 to ZT3, from ZT8 to ZT11, and from ZT20 to ZT23; ZT0 was the time when the light turned on). The relations of the real time points with the ZT time points were as follows: ZT3—3 h of stress started at 7 a.m. (ZT0; light on) and finished at 10 a.m.; ZT11—3 h of stress started at 3 p.m. (ZT8) and finished at 6 a.m.; ZT23—3 h of stress started the next day at 3 a.m. (ZT20) and finished at 6 p.m. (please see Figure 1). At the end of the experimental period, the control and stressed animals were quickly decapitated without anesthesia, and trunk blood was collected. Individual serum samples were stored at −80 °C until they were assayed for androgen (testosterone + dihydrotestosterone; T+DHT) and corticosterone (CORT) levels (Figure 2). In each experiment, the control and experimental group consisted of 12 to 18 animals randomly divided into three time-point groups, with 4 to 6 animals per time point. The sample size was checked by Power Analysis using G Power software (http://core.ecu.edu/psyc/wuenschk/Power.htm, accessed on 4 March 2022) according to previous results obtained by our group. The experiments were repeated two times. The experimental design is presented in Figure 1.

### 2.4. Serum Hormones Measurement

The level of androgens, in serum, was referred to as T+DHT considering that the anti-testosterone serum №250 showed 100% cross-reactivity with DHT (for references, please see [20,22]). Serum androgen levels were measured via radioimmunoassay. All samples were measured in duplicate in one assay (sensitivity: 6 pg per tube; intra-assay coefficient of variation 5–8%). Serum corticosterone (CORT) levels in all samples were measured in duplicate in one assay using the corticosterone EIA Kit (Cayman Chemical, Michigan, MI, USA) with 30 pg/mL as the lowest standard significantly different from blank.

### 2.5. Spermatozoa Isolation 

The isolation of caudal epididymides spermatozoa was carried out following the WHO laboratory manual (https://www.who.int/reproductivehealth/publications/infertility/9789241547789/en/, accessed on 4 March 2022) with modifications for rat spermatozoa isolation (for references, please see [21,22]). In short, caudal epididymides were quickly isolated, and surrounding adipose tissue was removed. Isolated epididymis was placed in a Petri dish containing 4 mL of the medium for the isolation and preservation of spermatozoa (1% M199 in HBSS with 20 mM HEPES buffer and 5% BSA) or Whitten’s Media (100 mM NaCl, 4.7 mM KCl, 1.2 mM KH_2_PO_4_, 1.2 mM MgSO_4_, 5.5 mM glucose, 1 mM pyruvic acid, and 4.8 mM lactic acid), depending on the subsequent analysis. Isolated epididymides were finely punctuated with a 25G needle to enable spermatozoa to be released into the medium and incubated at 37 °C for 10 min. Released spermatozoa were collected and centrifuged for 5 min at 700× *g* at room temperature. The supernatant was removed, and the pellet was resuspended in the appropriate medium depending on the subsequent analysis. Concentrations of isolated spermatozoa were calculated using a Makler counting chamber (Sefi-Medical Instruments, Ltd., Haifa, Israel). Isolated spermatozoa were used for the capacitation and acrosome reaction procedure, and the rest of the spermatozoa were stored at −80 °C, before RNA isolation and the subsequent gene transcription analysis.

### 2.6. Spermatozoa Functionality Assessment (Capacitation and Acrosome Reaction)

To determine the spermatozoa functionality, approximately 1.5×10^5^ spermatozoa in 50 μL of Whitten’s Media was mixed with 350 μL of WH+ media (Whitten’s Media supplemented with the 10 mg/mL BSA (Bovine Serum Albumin) and 20 mM of NaHCO_3_ to stimulate the capacitation) with a drop of mineral oil at 37 °C (5% CO_2_) for 1 h. An amount of 50 μL of capacitated spermatozoa was transferred in two new tubes, one without progesterone and one with 15 μM of progesterone (PROG), with a drop of mineral oil, and incubated at 37 °C (5% CO_2_) for 30 min. To activate the acrosome reaction, 15 μM of progesterone was added, while tubes without PROG were present as the control of the acrosome reaction. Following the stimulation of the acrosome reaction, 20 μL of the spermatozoa suspension from each tube was fixed with 100 μL of fixation solution (20 mM Na_2_HPO_4_, 150 mM NaCl, and 7.5% formaldehyde) for 20 min at room temperature. Subsequently, fixed spermatozoa were centrifuged for 1 min at 12000× *g* and washed with 100 mM ammonium acetate, pH 9. Smears of fixed spermatozoa were prepared on microscopic slides and air dried. Dried spermatozoa smears were stained using staining solution (0.04% Coomassie Blue—G250, 50% methanol, and 10% acetic acid) for 5 min at room temperature. The staining solution was rinsed with distilled water, and spermatozoa smears were allowed to air dry. Stained smears were analyzed using the microscope Leica DMLB 100T (Leica, Wetzlar, Germany), with 1000× magnification. Ten to fifteen photos per slide were taken with a Leica MC190 camera (Leica, Wetzlar, Germany) and LAS Ver 4.8.0 software, and up to 100 spermatozoa per slide were counted to determine the acrosomal status. Blue staining in the acrosomal region of the head indicated intact acrosome, while spermatozoa without blue staining in the acrosomal region were considered to be acrosome-reacted. Data are presented as the percentage of acrosome-reacted spermatozoa ± SEM.

### 2.7. Isolation of RNA and cDNA Synthesis

Spermatozoa samples isolated from caudal epididymides were stored at −80 °C until they were used for the isolation of total RNA. Total RNA isolation was performed using the GenElute™ Mammalian Total RNA Miniprep Kit following the protocol recommended by the manufacturer (Sigma Aldrich, Steinheim am Albuch, Baden-Wurttemberg, Germany). To eliminate DNA from the samples, DNase I (RNase-free) treatments were carried out according to the manufacturer’s instructions (New England Biolabs, Ipswich, MA, USA). The concentration and purity of isolated total RNA were measured using the BioSpec-nano spectrophotometer (Shimadzu, Kyoto, Japan). Furthermore, the first-strand cDNA was synthesized using the High-Capacity Kit for cDNA preparation following the manufacturer’s protocol (Thermo Fisher Scientific, Waltham, MA, USA). In each set of reactions, negative controls were included. The quality of RNA and DNA integrity were checked using control primers for *Gapdh*, as described previously by our group (for references, please see [21,22,36]).

### 2.8. Relative Quantification of Gene Expression

The relative expression of the genes was quantified by real-time PCR (RQ-PCR) using SYBR^®^ Green-based chemistry from Applied Biosystems (Thermo Fisher Scientific, Waltham, MA, USA). Each reaction contained 10 ng of cDNA (calculated from starting RNA) in the volume of 2.5 μL and specific primers at the final concentration of 500 nM. Primer sequences used for real-time PCR analysis and Ct values, as well as GenBank accession codes for full gene sequences (www.ncbi.nlm.nih.gov/sites/entrez, accessed on 4 March 2022), are given in Appendix A. Relative gene expression quantification of *Gapdh* was measured in each sample and used to correct the variations in cDNA content between samples. The relative quantification of each gene was performed in duplicate, three times for each independent in vivo experiment. The real-time PCR reactions were carried out in the Eppendorf Master Cycler ep RealPlex 4, and post-run analyses were performed using Mastercycler^®^ eprealplex Software 1.0 (for references, please see [21,22,36]).

### 2.9. Relative Quantification of Protein Expression

Rat spermatozoa samples isolated from caudal epididymides were frozen and stored at −80 °C until protein extraction. Cells were lysed, and Western blot analysis was performed as described previously [20]. Immune-reactive bands were detected using MyECL Imager (Thermo Fisher Scientific Inc.) and analyzed as two-dimensional images using Image J version 1.48 (http://rsbweb.nih.gov/ij/download.html, accessed on 4 March 2022). The optical density of images is expressed as volume adjusted for the background, which gives arbitrary units of adjusted volume. The normalization of the data was carried out using GAPDH protein expression as the endogenous control. Immune detection was performed with different antibodies (all details are listed in Appendix A). 

### 2.10. Statistical Analysis

The results of the experiments represent group means ± SEM values of the individual variation from two independent experiments (4 to 6 rats per group). Results from each experiment were analyzed using Mann–Whitney’s unpaired nonparametric two-tailed test between the IMO group and control group within the same time point. All the statistical analyses were carried out using GraphPad Prism 5.0 Software (GraphPad Software 287 Inc., La Jolla, CA, USA). In all cases, a *p*-value < 0.05 was considered to be statistically significant.

### 2.11. Principal Component Analysis

Principal component analysis (PCA) was performed with dudi.PCA function implemented in “ade4” package [37], on scaled and centered data matrix, within the R environment. We decided to retain the first two PCs based on eigen values and cumulative variation. In support of such a decision, we performed Horn’s parallel analysis for a PCA with the “paran” package, to adjust for finite sample bias in retaining components [38]. Biplots visualization was performed with “factoextra” package [39].

## 3. Results

To examine the connection between repeated IMO stress (immobilization stress, the most common stress in human society) at different times during 24 h, on the one hand, and male (sub/in) fertility, on the other hand, 3 h of IMO stress for 10 consecutive days (10×3hIMO) was applied on adult male rats. Repeated IMO stress was applied from ZT0 to ZT3, from ZT8 to ZT11, and from ZT20 to ZT23. ZT0 was the time when the lights turned on, and it corresponded to 7 a.m. in real time. The hormones (corticosterone and testosterone), the number and functionality of spermatozoa, as well as the transcriptional profiles of 22 mitochondrial dynamics and function markers and 22 signaling molecules, regulating both spermatozoa number/function and mitochondrial dynamics, were analyzed (Figure 1).

### 3.1. Repeated Stress Decline Spermatozoa Number and Functionality at All the Analyzed ZT Time Points (ZT3, ZT11, and ZT23) 

The repeated stress (10×3hIMO) was effective as a stressor since the elevated corticosterone level in ZT3-10×3hIMO was 9.9-fold, ZT11-10×3hIMO was 2.9-fold, and ZT23-10×3hIMO was 8.9-fold compared to the ZT-corresponding controls (Figure 2A, left panel). The circadian-like profile of serum androgens (T+DHT) was evident in groups of undisturbed (control) rats (Figure 2A, right panel). The reduced level of androgens after 10×3hIMO repeated stress was persistent at ZT3 (12.3-fold), ZT11 (24.3-fold), and ZT23 (2.0-fold) compared to the ZT-corresponding controls (Figure 2A, right panel). 

The number of spermatozoa (Figure 2B) declined in rats exposed to 10×3hIMO stress at all the analyzed ZT time points: ZT3-10×3hIMO was 1.7-fold, ZT11-10×3hIMO was 2.0-fold, and ZT23-10×3hIMO was 1.9-fold compared to the ZT-corresponding controls (Figure 2B). Additionally, the spermatozoa functionality decreased in all experimental groups (Figure 2C): ZT3-10×3hIMO was 3.5-fold, ZT11-10×3hIMO was 2.8-fold, and ZT23-10×3hIMO was 2.7-fold compared to the ZT-corresponding controls.

In search of the possible mechanism(s) beyond these effects, the transcriptional profile of mitochondrial dynamics markers and signaling molecules regulating both mitochondrial dynamics and spermatozoa number and functionality were analyzed. The results showed that 10×3hIMO stress at all of the ZT time points analyzed dramatically disturbed the expressions of transcripts for the markers of mitochondrial dynamics and functionality, as well as related signaling pathways in spermatozoa. The expression levels of 40 out of 44 (90.9%) transcripts were changed from the ZT-corresponding control at a particular ZT time point (Figure 3, Figure 4, Figure 5, Figure 6, Figure 7, Figure 8 and Figure 9, Table 1).

### 3.2. The Significant Changes in Transcriptional Profiles of Mitochondrial Dynamics and Functionality Markers in Spermatozoa from Repeatedly Stressed Rats Are Evident at All the Analyzed ZT Time Points (ZT3, ZT11, and ZT23) 

The transcriptional profiles of molecular markers of mitochondrial dynamics and functionality in spermatozoa were disturbed by 10×3hIMO stress, since the transcriptional levels of 20 out of 22 (90.9%) markers were changed (Figure 3, Figure 4, Figure 5, Figure 6 and Figure 7, Table 1).

Mitochondrial biogenesis markers changed in 7 out of 8 transcripts =>87.5%. The levels of transcripts for gene encoding PGC1 (*Ppargc1a*), very well known as the master regulator involved in the transcriptional control of all the processes related to mitochondrial homeostasis and the integrator of environmental signals [1,13] were disturbed (Figure 3A,C).

A circadian-like profile was observed in the expression of ***Ppargc1a*** transcript since it differently changed in spermatozoa taken from 10×3hIMO stressed rats: it increased in the ZT3-10×3hIMO group (3.1-fold compared to ZT3-Control) and in ZT23-10×3hIMO (2.6-fold compared to ZT23-Control), but unchanged in ZT11-10×3hIMO (compared to ZT11-Control). There were no effects of the 10×3hIMO stress observed on the transcription of ***Ppargc1b*** compared to the ZT-corresponding control. 

The transcription profiles of PGC1 downstream targets (*Nrf1*, *Nrf2a*, *Tfam*, *mtNd1*, and *Ppard*) that regulate the genes for subunits of oxidative phosphorylation (OXPHOS) also changed. 

***Tfam*** transcription increased at ZT3 and ZT23: in the ZT3-10×3hIMO group, it was 1.7-fold compared to ZT3-Control, and in the ZT23-10×3hIMO group, it was 1.6-fold compared to ZT23-Control (Figure 3E).

***Nrf1*** transcript increased in spermatozoa in the ZT3-10×3hIMO group 2.7-fold (compared to ZT3-Control) and in the ZT23-10×3hIMO group 3.1-fold (compared to ZT23-Control), but decreased in the ZT11-10×3hIMO group 3.0-fold (compared to ZT11-Control) (Figure 3B).

***Nrf2a*** transcription increased in spermatozoa in the ZT3-10×3hIMO group 1.8-fold and in the ZT23-10×3hIMO group 1.9-fold compared to the ZT-corresponding control. In the ZT11-10×3hIMO group, *Nrf2a* transcription decreased 1.6-fold compared to the ZT-corresponding control (Figure 3B).

The ***Ppara*** transcription profile only increased in spermatozoa obtained from rats in the ZT11-10×3hIMO group (2.4-fold), while in the ZT3-10×3hIMO and ZT23-10×3hIMO groups, it remained unchanged compared to the ZT-corresponding control (Figure 3F).

***Ppard*** transcription increased in spermatozoa obtained from rats in the ZT3-10×3hIMO (2.0-fold) and ZT23-10×3hIMO (1.7-fold) groups, while in the ZT11-10×3hIMO group, it decreased (2.4-fold) compared to the ZT-corresponding control (Figure 3F).

The ***mtNd1*** transcription profile only changed in ZT3-10×3hIMO (increased 5.7-fold compared to ZT3-Control), and in the ZT11-10×3hIMO and ZT23-10×3hIMO groups it remained unchanged compared to the ZT-corresponding control (Figure 3G).

Mitochondrial fusion markers in 3 out of 3 = >100%. The changes in the transcriptional profiles of all spermatozoal mitofusion as well as mito-architecture markers (*Mfn1*, *Mfn2*, and *Opa1*) were observed at all the ZT time points analyzed—ZT3, ZT11, and ZT23—and had the same circadian-like profile—significantly increased at ZT3 and ZT23 and significantly decreased at ZT11 (Figure 4). 

***Mfn1*** transcription significantly increased in spermatozoa obtained from rats in the ZT3-10×3hIMO (3.7-fold) and ZT23-10×3hIMO (1.9-fold) groups, while in the ZT11-10×3hIMO group, it decreased (2.3-fold) compared to the ZT-corresponding control.

The ***Mfn2*** transcription profile was similar to *Mfn1*. The level of *Mfn2* transcription decreased in spermatozoa from the ZT3-10×3hIMO (3.0-fold) and ZT23-10×3hIMO (1.5-fold) groups, while in the ZT11-10×3hIMO group, it decreased (3.3-fold) compared to the ZT-corresponding control.

The ***Opa1*** transcript profile was similar to *Mfn1* and *Mfn2*. The level of *Opa1* transcription increased in the ZT3-10×3hIMO (1.7-fold) and ZT23-10×3hIMO (2.1-fold) groups, while in the ZT11-10×3hIMO group, it decreased (1.9-fold) compared to the ZT-corresponding control.

Mitochondrial fission markers changed in 1 out of 2 = >50%. The level of transcripts for *Drp1* significantly changed since *Fis1* remained unchanged at different ZT time points (Figure 5).

The ***Drp1*** transcript profile (circadian-like, as that observed in mitochondrial fusion marker transcriptional analyses) significantly increased in spermatozoa obtained from rats in the ZT3-10×3hIMO (3.0-fold) and ZT23-10×3hIMO (4.2-fold) groups, while in the ZT11-10×3hIMO group, it decreased (2.0-fold) compared to the ZT-corresponding control.

Mitochondrial autophagy markers changed in 3 out of 3 = >100%. The significant changes were evident on the transcription profile of all of the mitochondrial autophagy markers analyzed (*Pink1*, *Prkn*, and *Tfeb*) and also showed a circadian-like pattern (Figure 6).

***Pink1*** significantly increased in spermatozoa from the rats in the ZT3-10×3hIMO (2.1-fold) and ZT23-10×3hIMO (3.8-fold) groups, while in the ZT11-10×3hIMO group, it decreased (2.4-fold) compared to the ZT-corresponding control.

***Prkn*** increased in the ZT3-10×3hIMO (2.1-fold) and ZT23-10×3hIMO (2.8-fold) groups, while in the ZT11-10×3hIMO group, it decreased (2.2-fold) compared to the ZT-corresponding control.

***Tfeb*** increased in the ZT3-10×3hIMO (3.1-fold) group and decreased in the ZT11-10×3hIMO group (3.0-fold) compared to the ZT-corresponding control. There was no significant change observed in the ZT23-10×3hIMO group.

Mitochondrial functionality markers changed in 6 out of 6 = >100%. The transcriptional profiles of NRF1/NRF2 downstream targets (CytC, COX4, UCPs) serving as mitochondrial functional markers as well as the mediators of regulated proton leak and controllers of the production of superoxide and other downstream reactive oxygen species [40] were significantly changed at different ZT time points but did not show any regular circadian pattern (Figure 7). 

***Cox4i1*** transcription significantly increased in spermatozoa from the ZT3-10×3hIMO group (3.2-fold vs. ZT3-Control) and decreased in the ZT11-10×3hIMO group (2.3-fold vs. ZT11-Control), while in the ZT23-10×3hIMO group, it remained unchanged (vs. ZT23-Control).

The ***Cox4i2*** transcription level significantly increased in spermatozoa from all experimental groups compared to the ZT-corresponding controls: ZT3-10×3hIMO (3.3-fold), ZT11-10×3hIMO (1.7-fold), and ZT23-10×3hIMO (1.9-fold).

***Cytc*** transcription significantly changed— it only increased in the ZT3-10×3hIMO group (1.7-fold vs. ZT3-Control), while the ***Ucp1*** transcription level was only significantly lower in the ZT11-10×3hIMO group (2.4-fold compared to ZT11-Control).

***Ucp2*** transcription in spermatozoa changed at all the ZT time points analyzed. The *Ucp2* transcription level increased in spermatozoa isolated from the ZT3-10×3hIMO (3.8-fold compared to ZT3-Control) and ZT23-10×3hIMO groups (1.5-fold compared to ZT23-Control) but decreased in spermatozoa from the ZT11-10×3hIMO group (3.9-fold compared to ZT11-Control).

***Ucp3*** transcription decreased in the ZT3-10×3hIMO group (5.8-fold compared to ZT3-Control) and increased in the ZT11-10×3hIMO group (1.5-fold compared to ZT11-Control), and in the ZT23-10×3hIMO group, it was unchanged compared to ZT23-Control.

The results of the PCA analysis show the significant separation of the effects of the repeated stress on mitochondrial dynamics markers depending on the day phase. It is clear that the transcriptional patterns were different during the inactive/light and active/dark phases of the rats’ day (Figure 12A).

It is important to point out that the transcriptional profiles of most of the mitochondrial dynamics and functionality markers in spermatozoa obtained from control rats showed circadian-like patterns. When controls at ZT3 were used as the calibrators, the circadian-like fashion of response was observed for *Ppargc1b*, *Nrf1*, *Nrf2a*, *Ppard* (Appendix A), *Mfn1*, *Mfn2*, *Opa1* (Appendix A), *Pink1*, and *Ucp2* (Appendix A), while others were not changed (Appendix A).

### 3.3. The Significant Changes in Transcriptional Profiles of Signaling Molecules Regulating the Number and Functionality of Spermatozoa, as Well as the Mitochondrial Dynamics and Functionality in Spermatozoa from Repeatedly Stressed Rats Are Also Evident at All the Analyzed ZT Time Points (ZT3, ZT11, ZT23)

Since cAMP and MAPK signaling are crucial not only for the regulation of spermatozoa number and functionality [41], but also for the regulation of mitochondrial dynamics and functionality [1,13,42], the transcriptional profiles of the main signaling molecules were analyzed. The markers of these signaling pathways significantly changed at all of the ZT time points analyzed (ZT3, ZT11, and ZT23). The transcriptional level of 20 out of 22 (91%) markers changed, and most of them had a circadian-like pattern of expression (Figure 8 and Figure 9, Table 1). 

cAMP signaling markers changed in 11 out of 12 = >92%. As a consequence of 10×3hIMO stress at different ZT time points, the transcriptional profile of almost all of the cAMP signaling markers analyzed changed, including adenylyl cyclases (*Adcy3*, *Adcy5*, *Adcy6*, *Adcy7*, *Adcy8*, and *Adcy9*), except *Adcy10*, and protein kinase A subunits (*Prkaca*, *Prkacb*, *Prkar1a*, *Prkar2a*, and *Prkar2b*). Oppositely to the adenylyl cyclases, all of the catalytic and regulatory protein kinase A subunits analyzed shared a very similar circadian-like pattern of expression (Figure 8).

***Adcy3*** transcription levels significantly changed—they only decreased in the ZT11-10×3hIMO group (3.8-fold vs. ZT11-Control) and remained unchanged in the other two groups.

The ***Adcy5*** transcriptional profile changed (increased) in spermatozoa from the ZT11-10×3hIMO (1.6-fold vs. ZT11-Control) and ZT23-10×3hIMO (1.5-fold vs. ZT23-Control) groups. In spermatozoa from the ZT3-10×3hIMO group, there were no changes (vs. ZT3-Control).

***Adcy6*** transcription increased in spermatozoa isolated from the ZT3-10×3hIMO (3.6-fold vs. ZT3-Control) and ZT23-10×3hIMO (2.1-fold vs. ZT23-Control) groups.

***Adcy7*** transcription levels increased in spermatozoa from the ZT23-10×3hIMO group (1.5-fold vs. ZT23-Control) and decreased in spermatozoa from the ZT11-10×3hIMO group (2.7-fold vs. ZT11-Control). 

***Adcy8*** transcription increased in spermatozoa from the ZT11-10×3hIMO group (3.6-fold vs. ZT11-Control) and decreased in spermatozoa from the ZT3-10×3hIMO group (2.9-fold vs. ZT3-Control). 

The ***Adcy9*** transcriptional profile changed in the opposite way to ***Adcy8***: it increased in spermatozoa from the ZT3-10×3hIMO (3.0-fold vs. ZT3-Control) group and decreased in spermatozoa from the ZT11-10×3hIMO (1.8-fold vs. ZT11-Control) group. 

***Prkaca*** transcription increased in spermatozoa from the ZT3-10×3hIMO (1.9-fold vs. ZT3-Control) group and decreased in spermatozoa from the ZT11-10×3hIMO (2.9-fold vs. ZT11-Control) group. 

The ***Prkacb*** transcription profile changed at all of the ZT time points analyzed: the transcript level increased in spermatozoa isolated from the ZT3-10×3hIMO group (3.1-fold compared to ZT3-Control) and ZT23-10×3hIMO group (1.8-fold compared to ZT23-Control) but decreased in spermatozoa from the ZT11-10×3hIMO group (3.6-fold compared to ZT11-Control).

***Prkar1a*** transcription also changed at all of the ZT time points analyzed: the transcript level increased in spermatozoa isolated from the ZT3-10×3hIMO group (2.6-fold compared to ZT3-Control) and ZT23-10×3hIMO group (1.7-fold compared to ZT23-Control) but decreased in spermatozoa from the ZT11-10×3hIMO group (3.3-fold compared to ZT11-Control).

The ***Prkar2a*** transcriptional profile was similar to the profiles of the previous transcripts for catalytic/regulatory subunits of PRKA: it increased in spermatozoa isolated from the ZT3-10×3hIMO (2.1-fold compared to ZT3-Control) and ZT23-10×3hIMO groups (1.4-fold compared to ZT23-Control) but decreased in spermatozoa from the ZT11-10×3hIMO group (1.8-fold compared to ZT11-Control).

The ***Prkar2b*** transcription profile was similar to *Prkar2a.* The level of *Prkar2b* transcripts increased in spermatozoa isolated from the ZT3-10×3hIMO (3.1-fold compared to ZT3-Control) and ZT23-10×3hIMO groups (1.8-fold compared to ZT23-Control) but did not change in spermatozoa from the ZT11-10×3hIMO group (compared to ZT11-Control).

The results of PCA analysis show the significant separation of the effects of the repeated stress on cAMP signaling pathway elements depending on the day phase. It is clear that the transcriptional patterns were different during the inactive and active phases (Figure 12B).

MAPK signaling markers changed in 9 out of 10 = >90%. All of the markers of MAPK signaling analyzed (*Mapk1*, *Mapk3*, *Mapk6*, *Mapk8*, *Mapk9*, *Mapk11*, *Mapk12*, *Mapk13*, and *Mapk14*), except *Mapk7*, were affected at different ZT time points (Figure 9).

***Mapk1*** transcription increased in spermatozoa isolated from the ZT3-10×3hIMO group (4.0-fold compared to ZT3-Control) and decreased in spermatozoa from the ZT11-10×3hIMO group (2.8-fold compared to ZT11-Control). The transcription of *Mapk1* did not significantly change in spermatozoa from the ZT23-10×3hIMO group (to ZT23-Control).

***Mapk3*** transcription increased in spermatozoa from both the ZT3-10×3hIMO (2.0-fold compared ZT3-Control) and ZT23-10×3hIMO (1.5-fold compared to ZT23-Control) groups.

The ***Mapk6*** transcriptional profile, like the profile of *Mapk3*, increased in spermatozoa from both ZT3-10×3hIMO (1.8-fold compared ZT3-Control) and ZT23-10×3hIMO (2.0-fold compared to ZT23-Control) groups without significant changes in the ZT11-10×3hIMO group (compared to ZT11-Control).

The ***Mapk8*** and ***Mapk9*** transcriptional profiles were similar to *Mapk1*. *Mapk8* increased in spermatozoa isolated from the ZT3-10×3hIMO group (3.8-fold compared to ZT3-Control) and decreased in spermatozoa from the ZT11-10×3hIMO group (2.0-fold compared to ZT11-Control). *Mapk9* increased in spermatozoa isolated from the ZT3-10×3hIMO group (2.1-fold compared to ZT3-Control) and decreased in spermatozoa from the ZT11-10×3hIMO group (1.9-fold compared to ZT11-Control). There was no significant change in the ZT23-10×3hIMO group (vs. ZT-corresponding control) for both genes.

The ***Mapk11*** transcriptional profile decreased in spermatozoa from the ZT3-10×3hIMO (1.8-fold compared ZT3-Control) group and increased in spermatozoa from the ZT23-10×3hIMO (1.5-fold compared to ZT23-Control) group without a significant change in the ZT11-10×3hIMO group (compared to ZT11-Control).

***Mapk12*** transcription significantly changed at all of the ZT time points analyzed: the transcription level increased in spermatozoa isolated from the ZT3-10×3hIMO (2.8-fold compared to ZT3-Control) and ZT23-10×3hIMO groups (2.0-fold compared to ZT23-Control) but decreased in spermatozoa from the ZT11-10×3hIMO (3.7-fold compared to ZT11-Control) group.

***Mapk13*** transcription increased in spermatozoa from both ZT3-10×3hIMO (2.7-fold compared to ZT3-Control) and ZT23-10×3hIMO (2.5-fold compared to ZT23-Control) groups, without a significant change in the ZT11-10×3hIMO group (compared to ZT11-Control). 

***Mapk14*** transcription, like *Mapk1*, *Mapk 8* and *Mapk 9*, increased in the ZT3-10×3hIMO group (2.2-fold compared to ZT3-Control) and decreased in the ZT11-10×3hIMO group (2.9-fold compared to ZT11-Control), without a significant change in the ZT23-10×3hIMO group (compared to ZT23-Control).

The results of PCA analysis show the significant separation of the effects of the repeated stress on MAPK signaling pathway elements depending on the day phase. It is clear that the transcriptional patterns were different during the inactive and the active phases and that transcripts for the most important spermatozoal MAPK, MAPK11/p38MAPK, was highly expressed during the active phase (Figure 12C).

When controls at ZT3 were used as the calibrators, the circadian-like patterns were observed for *Adcy8* and *Adcy9* (Appendix A), as well as *Mapk3*, *Mapk7*, *Mapk12*, and *Mapk14* (Appendix A).

### 3.4. Repeated IMO Stress Resulted in a Circadian Transcriptional Pattern of the Majority of Analyzed Mitochondrial Dynamics/Functionality Markers, as Well as cAMP and MAPK Signaling-Pathway-Related Molecules, in Spermatozoa, Suggesting an Adaptive Response

Analyses have shown different patterns of transcription at different time points in spermatozoa after repeated immobilization stress. The transcriptional profiles of 91% (20/22) of mitochondrial dynamics and functionality markers and 86% (19/22) of signaling molecules were disturbed after repeated stress. Similar molecular changes in transcriptional profiles were observed at ZT3 and ZT23, but the opposite was observed at ZT11, suggesting the circadian nature of an adaptive response (Figure 10, Figure 11 and Figure 12, Table 1).

All of the analyzed markers of mitochondrial dynamics and functionality (except *Ppargc1b* and *Fis1*) were affected at different ZT time points. The majority of them (*Ppargc1a*, *Tfam*, *Nrf1*, *Ppard*, *Mfn1*, *Mfn2*, *Opa1*, *Drp1*, *Pink1*, *Prkn*, and *Ucp2*) had a circadian-like pattern of transcription, which implies significantly increased expression at ZT3, significantly (or insignificantly) decreased expression at ZT11, and significantly increased expression at ZT23. Most exceptions from this circadian pattern occurred among the mitochondrial functionality markers, while the transcription of mitofusion/architecture markers was the most consistent followed by the highest fold of change (Figure 10A, Figure 11A and Figure 12A).

Transcriptional analyses of cAMP signaling pathway molecules showed high heterogeneity at different ZT time points: only *Adcy6* had a circadian-like pattern similar to that described above. All of the analyzed protein kinase A subunits (*Prkacb*, *Prkar1a*, *Prkar2a*, and *Prkar2b*), except *Prkaca*, had the same time-dependent profile (increased transcription at ZT3 and ZT23). Most of the followed markers of the MAPK signaling pathway (*Mapk1*, *Mapk3*, *Mapk6*, *Mapk8*, *Mapk9*, *Mapk12*, *Mapk13*, and *Mapk14*) had a similar circadian-like pattern of transcription, which is also represented by increased expression at ZT3, decreased expression at ZT11, and increased (or not changed—*Mapk1*, *Mapk8*, *Mapk9*, and *Mapk14*) expression at ZT23 (Figure 10B, Figure 11B and Figure 12B,C).

## 4. Discussion

Mitochondria, with a wide variety of functions and the ability of intercellular trafficking, are crucial for the homeostasis of all cells [1,2], but especially for those with high energy demands such as spermatozoa [17]. Additionally, mitochondria are very important and are involved in first-line stress response [23]. Many studies discussed the correlation between stress and/or a stressful life (such as shift work) and male (in/sub)fertility. Yet, the mechanisms have not been described.

Here, for the first time, an in vivo approach was designed to mimic situations in which the human population is exposed to repeated psychological stress (the most common stress in human society) at different time points (ZT3, ZT11, and ZT23) during the day (24 h). Three time points were chosen (two points during 12 h light/inactive phase and one point during 12 h dark/active phase to serve as the “situation of shift work”). The results show, to the best of our knowledge for the first time, that repeated stress induced a physiological stress response in spermatozoa, illustrated by a circadian-like-pattern in the transcriptional profiles of mitochondrial dynamics and functionality markers and signaling molecules regulating both mitochondrial dynamics and spermatozoa number and functionality. The circadian nature of the spermatozoal adaptive response is, at least in part, driven by rhythmic corticosterone and testosterone secretion. Additionally, it is noteworthy to mention that the transcriptional profiles of most of the mitochondrial dynamics and functionality markers and some of the regulatory proteins in spermatozoa obtained from control rats showed circadian-like patterns. 

As was expected and shown by our group [43], corticosterone and testosterone secretions have a rhythmic pattern. These results are not new, but we wanted to show them to prove the accuracy of the model. It is well known that corticosterone acts as a synchronizer of a peripheral clock [44]. Thus, corticosterone together with testosterone could be involved in the circadian nature of the regulation of molecular events in spermatozoa. Our previous results showed that the stress hormone adrenaline changes mitochondrial functionality and markers with consequences on spermatozoa functionality using adrenergic signaling [21,22]. Here, we show that in situations where psychological stress is present at different time points during the day, the spermatozoal transcriptional profiles of most mitochondrial dynamics and functionality markers as well as most signaling molecules regulating both mitochondrial dynamics and spermatozoa number/functionality are similar at ZT3 and ZT23, but opposite at ZT11 (Figure 10, Figure 11 and Figure 13). 

The decreased number and functionality of epididymal spermatozoa were registered at all time points. According to the best of our knowledge, there are no published pieces of evidence regarding the effects of stress on spermatozoa number and functionality at different time points during the day. However, the results obtained at ZT3 are in line with findings showing that chronic intermittent stress decreases the number of spermatozoa [22,27], the number of spermatogenic cells [45], as well as spermatozoa motility [46] and sperm quality [47] in male rats. This could be the consequence of the inhibitory role of adrenaline on spermatozoa functionality [21] and/or the impairment of spermatogenesis as a result of the activation of stress-induced GRs signaling [48]. In humans, reduced levels of testosterone and spermatozoa motility was reported in high and moderate male runners [49]. Additionally, stress was shown to induce a decline in progressively motile spermatozoa [50], while in patients with post-traumatic stress disorder, higher secondary infertility was registered [51]. All stress signaling molecules are very well known as essential regulators of spermatozoa number and functionality [41] but also as important regulators of mitochondrial dynamics [1,13,42]. Accordingly, all are very important for (in)fertility.

Heat map analysis (Figure 10A) and patterns (Figure 11A) of the transcriptional profile of mitochondrial dynamics and functionality markers in spermatozoa obtained from adult rats repeatedly stressed at three time points during the day clearly show (Figure 13) that the transcription of most of the markers significantly increased (17 out of 22) at ZT3 (3 h after stress). A decreased level of the transcript was only registered for *Ucp3*. The changes were less pronounced at ZT11 (three increased, nine decreased) and ZT23 (10 increased). The most prominent circadian-like-patterns were registered for all main markers of mitochondrial fusion (*Mfn1*, *Mfn2*, and *Opa1*) and mitophagy (*Pink1* and *Prkn*), as well as some of the mitofission (*Drp1*) and the mitochondrial biogenesis (*Ppargc1a*, *Tfam*, and *Nrf1*) markers. 

It is difficult to compare our findings since published evidence is not available. The transcription of the genes is a multi-regulated process involving a plethora of signaling pathways directed by neuronal, endocrine, paracrine, autocrine, cryptocrine, and juxtacrine signals. In addition, the central and peripheral circadian clock systems interact with many different signals to produce an integrated output over the diurnal cycle, directing the cyclic activities in the cell [52]. Moreover, the existence and the role of the clock genes in spermatozoa are not clear and should be targets of future investigation. However, there is published evidence related to the effect of repeated stress at ZT3 from others and our group, but this has already been discussed in our previous publications [21,22]. The most interesting markers are obviously markers of mitofusion showing high similarity. This could be a target for the development of a new diagnostic toolkit since it has been shown that the expression level of MFN2 positively correlates with the motility and cryoprotective potential of human sperm [16], as well as with the mitofusin-mediated stimulation of OXPHOS [53]. The circadian-like pattern of *Tfam* transcription in spermatozoa presented here is also important for future clarification since TFAM gene expression positively correlates with abnormal forms, sperm DNA fragmentation, and mtDNA copy number [54,55]. A good circadian-like transcription pattern was observed for *Ucp2* and could be connected with the findings of other authors showing that the presence of UCP2 mitigates the loss of human spermatozoa motility [56].

Our findings provide new insights into the understanding of molecular events related to the possible circadian-like effects of repeated stress on spermatozoa, showing that 86% of markers of signaling pathways regulating both mitochondrial dynamics and spermatozoa number/functionality change during repeated stress. Again, heat map analysis (Figure 10B) and patterns (Figure 11B) of the transcriptional profile of signaling molecules regulating mitochondrial dynamics and functionality in spermatozoa obtained from rats repeatedly stressed at three time points during the day clearly show (Figure 13) that the transcription of most of the markers significantly increased (7 out of 12 cAMP signaling markers; 8 out of 10 MAPK signaling markers) at ZT3 (3 h after stress). At ZT3, the opposite pattern was observed: only levels of transcripts for *Adcy5* and *Adcy8* declined, while changes were registered in all 11 of the other markers. Again, the effect was less prominent at ZT11 (11 increased, 2 decreased) and ZT23 (12 increased) and again, at ZT23, only increases in the levels of transcription were observed. Compared to other *Adcy* isoforms, the opposite circadian-like pattern was registered for *Adcy8*. Additionally, the effects were more prominent on MAPK signaling markers. According to the best of our knowledge, there is no published evidence related to the circadian nature of the regulation of cAMP signaling and/or MAPK signaling in spermatozoa. However, all mentioned signaling molecules are very well known essential regulators of spermatozoa number/functionality [41], as well as regulators of PGC1, the biogenesis of OXPHOS, mitofusion, mitofission, and mitophagy [1,13,42]. Besides, all affected molecules are part of the complex signaling network in spermatozoa precisely regulated to provide fertility homeostasis in health and diseases [57]. The physiological meaning and the consequences of the increased expression of transcripts could be to keep basic spermatozoa functionality, since it was shown that cAMP signaling improves sperm motility [58,59] and it is important for the activation of CatSper channels [60]. Increased expressions of transcripts for all subunits of PRKA are also great adaptive and ameliorative mechanisms, since it was reported that PRKAR2A reduction in asthenozoospermic patients decreases sperm quality [61], while Prkar2B is sensitive to heat [62]. Last, but not least, increased transcripts for MAPK signaling markers could be compared with findings that testicular hyperthermia induces both MAPK1/3 and MAPK14 [63] and that MEK1/2 and ERK2 regulate the spermatozoa capacitation [64].

Figure 10, Figure 11, Figure 12 and Figure 13, as well as displaying expressions of PGC1 protein (the master regulator of mitochondrial dynamics and integrator of the environmental signals) and its down-stream target NRF2, clearly suggest the circadian nature of the response. We did not provide a molecular mechanism(s) nor insights into the molecular mechanism connecting the different transcriptional profiles to specific aspects related to mitochondrial dynamics and functionality. However, our results are (to the best of our knowledge) the first results showing the transcriptional profile of essential molecular markers of spermatozoa homeostasis and functionality after the application of repeated stress at different circadian time points during inactive/light and active/dark phases of the rats’ day. Our goal was not to search for the mechanism since we did not find anything related to the subject of our study in the published literature. The results clearly show that the transcriptional patterns of the main markers of spermatozoa have a circadian type of adaptive response. Moreover, in order to increase interpretability but at the same time minimize information loss, the results of PCA analysis (please see Figure 12) show the significant separation of the effects of repeated stress on mitochondrial dynamics markers, cAMP signaling pathway elements, and MAPK signaling pathway elements depending on the day phase. PCA confirmed that the transcriptional patterns were different during active and inactive phases with clear “separation” of *Adcy10* and *Mapk11/p38Mapk*, the main functional markers. In addition, p38MAPK protein expression exhibited a circadian-like profile.

Lastly, we believe that our results have a significant translational aspect related to the effect of stress at different time points and male fertility since, unfortunately, many recent publications reported an increased number of unexplained cases of infertility in men together with decreased fertility rates in men younger than 30 [17,18,65]. Those with stressful lives, i.e., alpha males (“life at the top”), exhibited significantly higher stress hormone levels than second-ranking-beta males [66]. Additionally, it is very well known that semen quality and fertility are important as fundamental markers of not only reproductive health but as the fundamental biomarkers of overall health [17,67]. Last, but not least, the World Health Organization (WHO) stated that the overall burden of infertility in men is high, unknown, underestimated, and has not displayed any decrease over the last 20 years. WHO called for urgent investigations of the mechanisms of fertility (https://www.who.int/reproductivehealth/topics/infertility/perspective/en/, accessed on 4 March 2022). In line with these facts are our preliminary results (Tomanic et al., unpublished results) showing that human spermatozoa samples from men with the same types of spermiograms but different stress levels express different transcriptional profiles of mitochondrial dynamics markers.

The limitation of this study is that RQ-PCR results were not correlated (due to technical reasons) with mitochondrial parameters that were obtained on “live” spermatozoa, but this was not the aim of our study. Additionally, it is important to keep in mind that changes in the light regime could have affected mitochondrial activity [68].

## 5. Conclusions

Repeated psychological stress applied at three time points during the day significantly decreased the number and functionality of spermatozoa at all time points. In the same samples, the transcriptional profiles of 91% (20/22) of mitochondrial dynamics and functionality markers and 86% (19/22) of signaling molecules were disturbed in a circadian-like manner. Similar molecular changes in transcriptional profiles were observed at ZT3 and ZT23, but the opposite was observed at ZT11, suggesting the circadian nature of the adaptive response. The results of PCA analysis show the significant separation of repeated stress effects during the inactive/light and active/dark phases of the day, suggesting the circadian timing of molecular adaptations. Accordingly, mitochondrial dynamics markers and signaling molecules regulating both mitochondrial dynamics and spermatozoa number and functionality are circadian-like adaptive mechanisms regulated by physiological stress response signaling.

## Figures and Tables

**Figure 1 cells-11-00993-f001:**
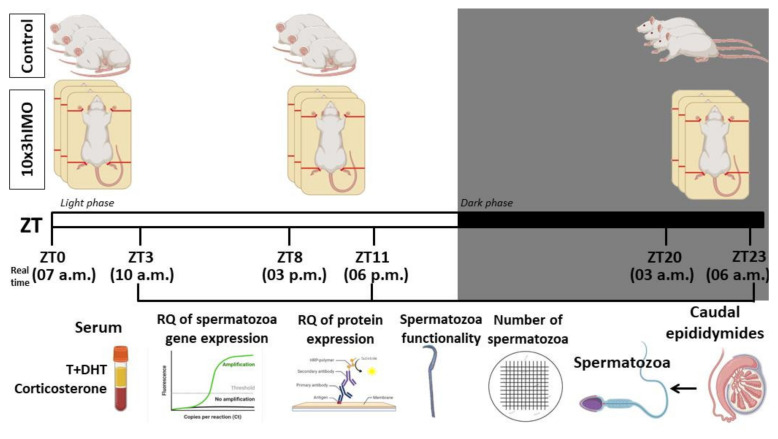
Experimental design of the in vivo experiment mimicking repeated stress applied at different time points during the day. Three-hour immobilization (IMO) stress was applied for 10 consecutive days (10×3hIMO) at different time points during the day (from ZT0 to ZT3, from ZT8 to ZT11, and from ZT20 to ZT23; ZT0 is the time when the light turned on). The relations of real time points with the ZT-time points: ZT3—3 h of stress started at 7 a.m. (ZT0; light on) and finished at 10 a.m.; ZT11—3 h of stress started at 3 p.m. (ZT8) and finished at 6 p.m.; ZT23—3 h of stress started next day at 3 a.m. (ZT20) and finished at 6 p.m. The levels of hormones, spermatozoa number and functionality (% acrosome reaction), as well as mitochondrial dynamics markers and related signaling molecules expressional profiles were followed. ZT—zeitgeber (time giver).

**Figure 2 cells-11-00993-f002:**
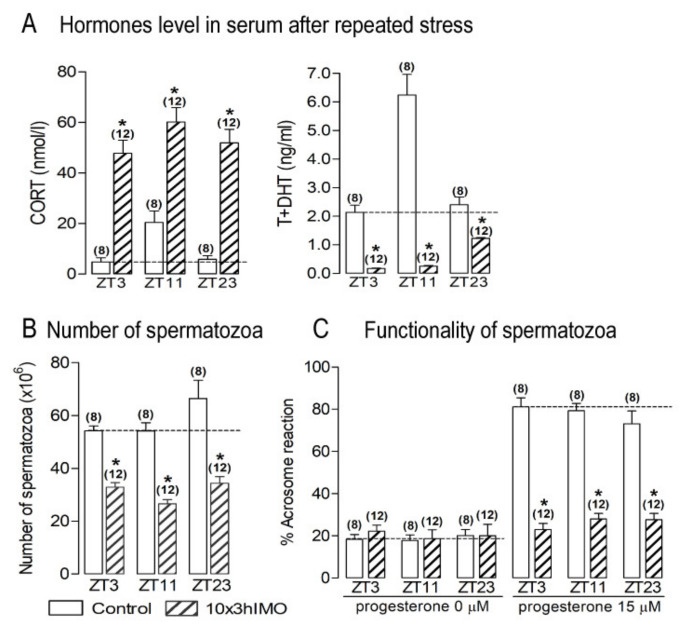
Repeated stress by immobilization (10×3hIMO) decreased androgen levels in circulation, together with functionality and number of spermatozoa, but increased the level of stress hormone corticosterone in different time points. Repeated psychophysical stress by immobilization (10×3hIMO) increased the circulating (**A**) stress hormone corticosterone level but decreased the level of androgens (testosterone + dihydrotestosterone, T+DHT). (**B**) The number of isolated spermatozoa from the caudal epididymides of undisturbed (control) rats and rats subjected to repeated immobilization stress, for 3 h for 10 consecutive days (10×3hIMO) in different periods during 24 h (from ZT0 to ZT3, from ZT8 to ZT11, and from ZT20 to ZT23; ZT0 was the time when the light turned on). (**C**) Spermatozoa functionality, presented as a % of acrosome-reacted spermatozoa, isolated from unstressed and repeatedly stressed (10×3hIMO) rats. After the capacitation, spermatozoa were stimulated with progesterone (PROG 15 µM) together with spermatozoa that were not treated with progesterone (PROG 0 µM). Acrosome-reacted spermatozoa were observed as the spermatozoa without the blue staining in the acrosome region, while blue staining in the acrosome region of the head of spermatozoa indicated intact acrosome. Data bars are mean ± SEM values of two independent in vivo experiments (n = number of rats). Statistical significance was set at level *p* < 0.05: * vs. control group of the same time point.

**Figure 3 cells-11-00993-f003:**
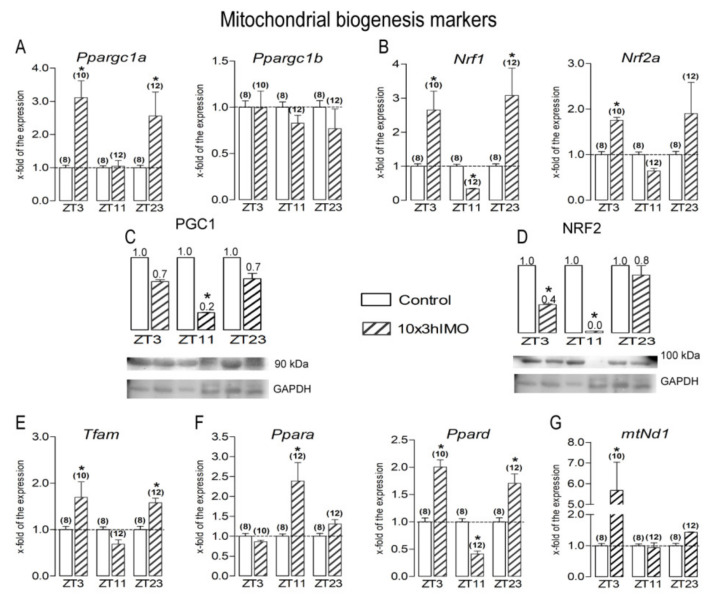
Different transcriptional profiles of mitochondrial biogenesis markers in spermatozoa of repeatedly stressed adult rats at different time points. Spermatozoa isolated from undisturbed and repeatedly stressed rats were used for RNA and protein isolation and further analysis of the transcriptional profile and protein expression profile of markers of mitochondrial biogenesis. The representative blots are shown as panels. Data from scanning densitometry were normalized to GAPDH (endogenous control). Values are shown as bars above the photos of blots, and numbers above the bars present fold of change. Data bars are mean ± SEM values of two independent in vivo experiments (n = number of rats). Statistical significance was set at level *p* < 0.05: * vs. control group of the same time point.

**Figure 4 cells-11-00993-f004:**
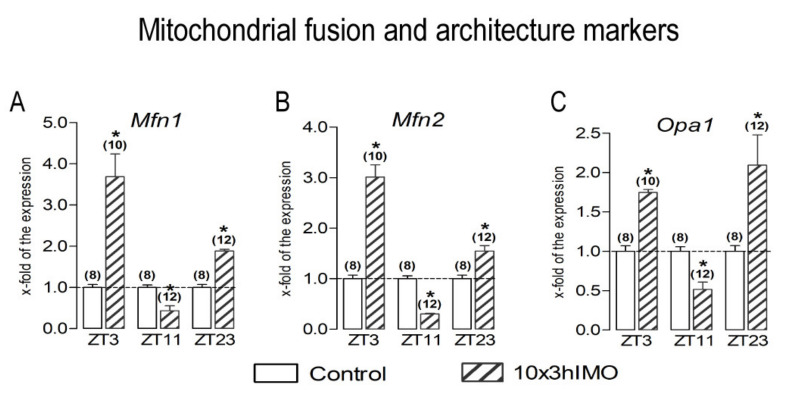
Different transcriptional profiles of mitochondrial fusion and architecture markers in spermatozoa of repeatedly stressed adult rats at different time points. Spermatozoa isolated from undisturbed and repeatedly stressed rats were used for RNA isolation and further analysis of the transcriptional profile of markers of mitochondrial fusion and architecture. Data bars are mean ± SEM values of two independent in vivo experiments (n = number of rats). Statistical significance was set at level *p* < 0.05: * vs. control group of the same time point.

**Figure 5 cells-11-00993-f005:**
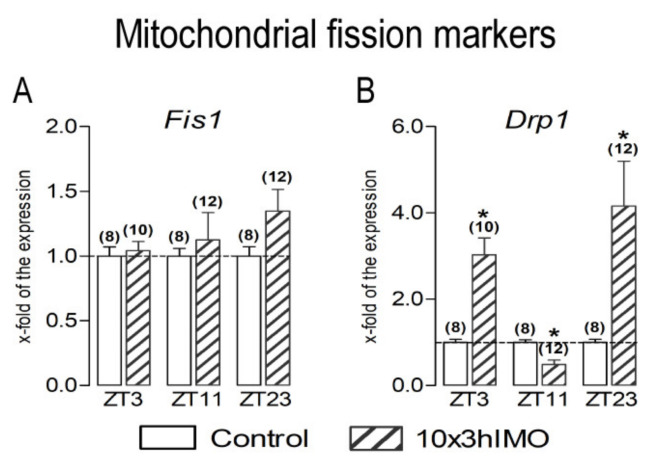
Different transcriptional profiles of mitochondrial fission markers in spermatozoa of repeatedly stressed adult rats at different time points. Spermatozoa isolated from undisturbed and repeatedly stressed rats were used for RNA isolation and further analysis of the transcriptional profile of markers of mitochondrial fission. Data bars are mean ± SEM values of two independent in vivo experiments (n = number of rats). Statistical significance was set at level *p* < 0.05: * vs. control group of the same time point.

**Figure 6 cells-11-00993-f006:**
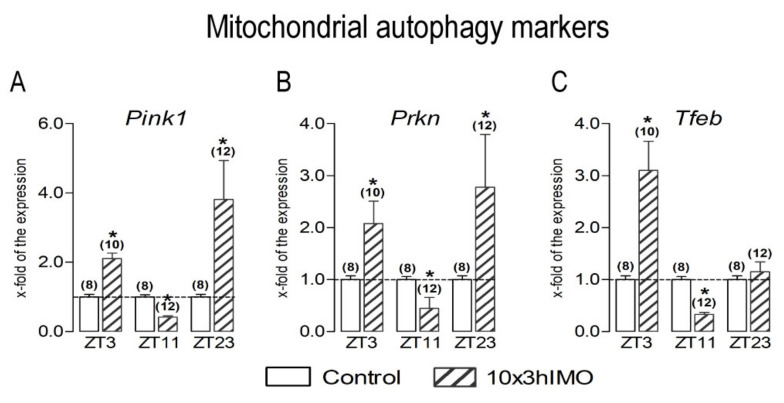
Different transcriptional profiles of mitochondrial autophagy markers in spermatozoa of repeatedly stressed adult rats at different time points. Spermatozoa isolated from undisturbed and repeatedly stressed rats were used for RNA isolation and further analysis of the transcriptional profile of markers of mitochondrial autophagy. Data bars are mean ± SEM values of two independent in vivo experiments (n = number of rats). Statistical significance was set at level *p* < 0.05: * vs. control group of the same time point.

**Figure 7 cells-11-00993-f007:**
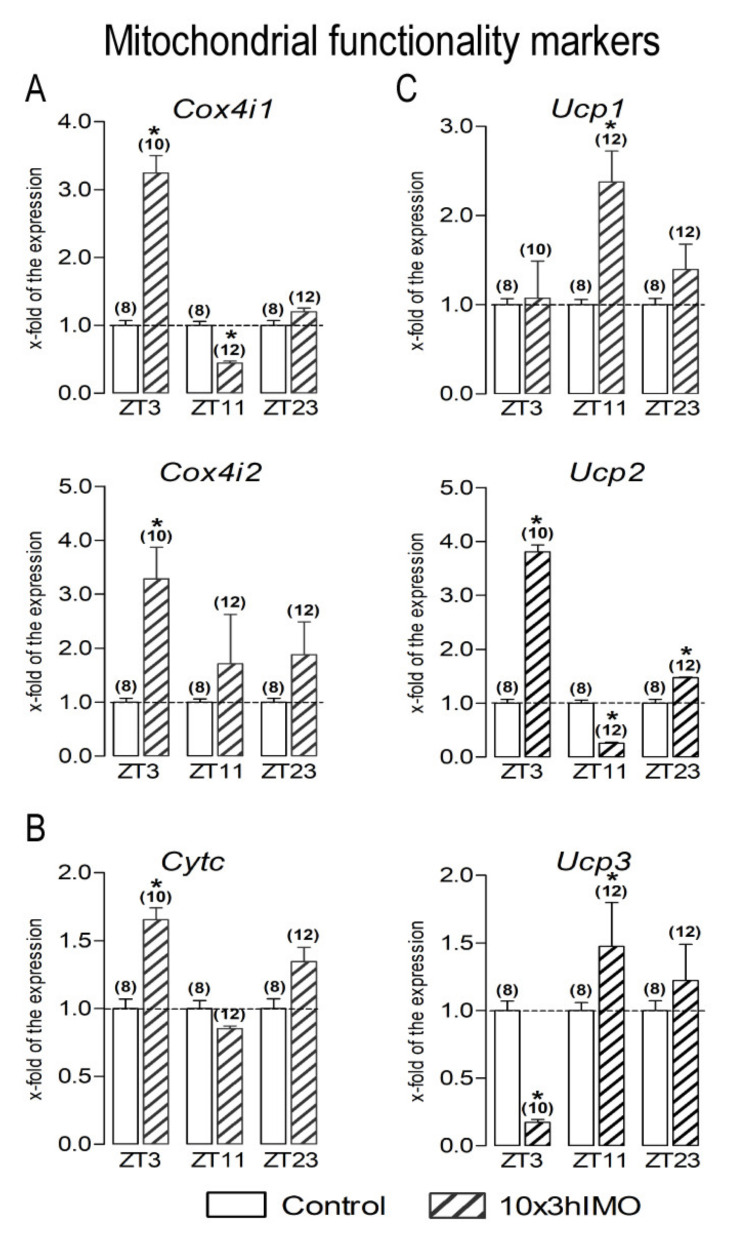
Different transcriptional profiles of mitochondrial functionality markers in spermatozoa of repeatedly stressed adult rats at different time points. Spermatozoa isolated from undisturbed and repeatedly stressed rats were used for RNA isolation and further analysis of the transcriptional profile of markers of mitochondrial functionality. Data bars are mean ± SEM values of two independent in vivo experiments (n = number of rats). Statistical significance was set at level *p* < 0.05: * vs. control group of the same time point.

**Figure 8 cells-11-00993-f008:**
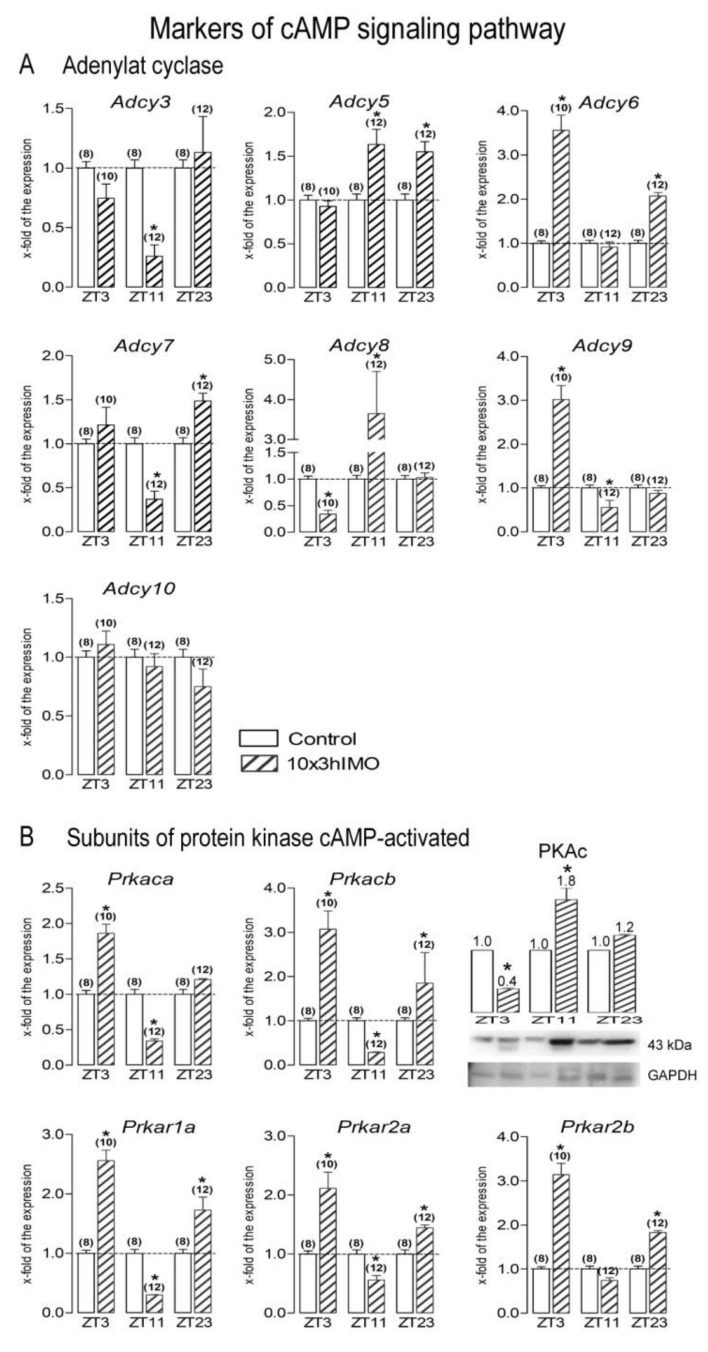
Different transcriptional profiles of markers of cAMP signaling regulating mitochondrial dynamics and functionality as well as spermatozoa number and functionality in spermatozoa of repeatedly stressed adult rats at different time points. Spermatozoa isolated from undisturbed and repeatedly stressed rats were used for RNA isolation and further analysis of the transcriptional profile of markers of cAMP signaling pathway. Data bars are mean ± SEM values of two independent in vivo experiments (n = number of rats). Statistical significance was set at level *p* < 0.05: * vs. control group of the same time point.

**Figure 9 cells-11-00993-f009:**
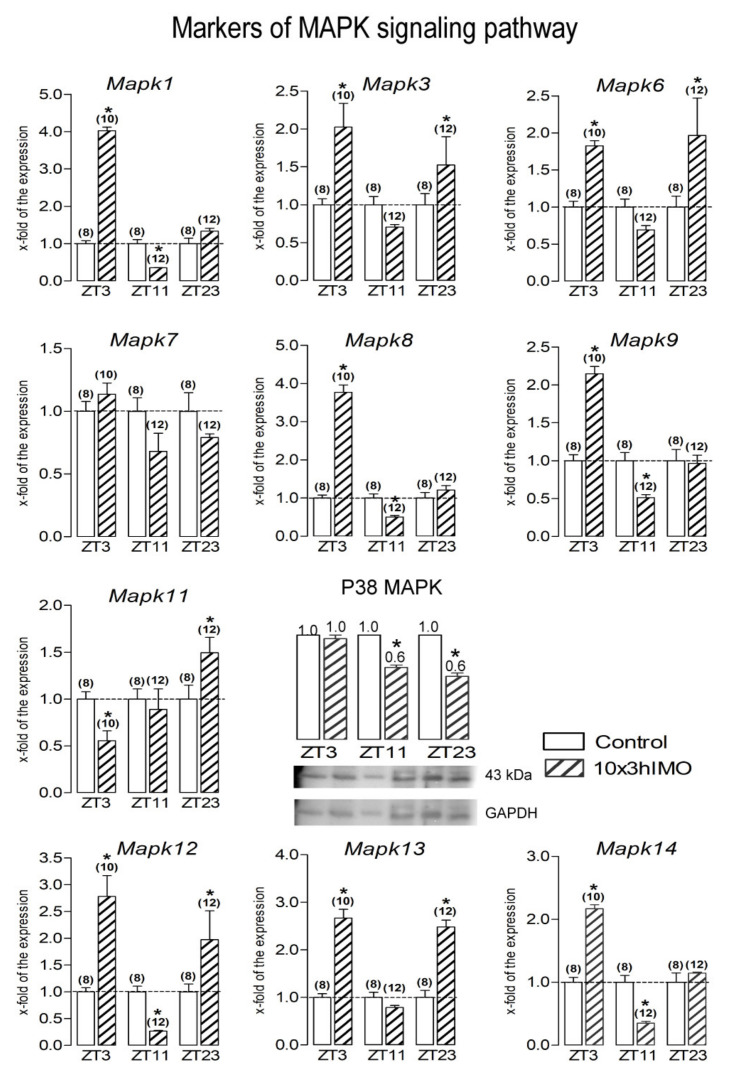
Different transcriptional profiles of markers of MAPK signaling regulating mitochondrial dynamics and functionality as well as spermatozoa number and functionality in spermatozoa of repeatedly stressed adult rats at different time points. Spermatozoa isolated from undisturbed and repeatedly stressed rats were used for RNA and protein isolation and further analysis of the transcriptional profile and protein expression profile of markers of MAPK signaling pathway. The representative blots are shown as panels. Data from scanning densitometry were normalized to GAPDH (endogenous control). Values are shown as bars above the photos of blots and numbers above the bars present fold of change. Data bars are mean ± SEM values of two independent in vivo experiments (n = number of rats). Statistical significance was set at level *p* < 0.05: * vs. control group of the same time point.

**Figure 10 cells-11-00993-f010:**
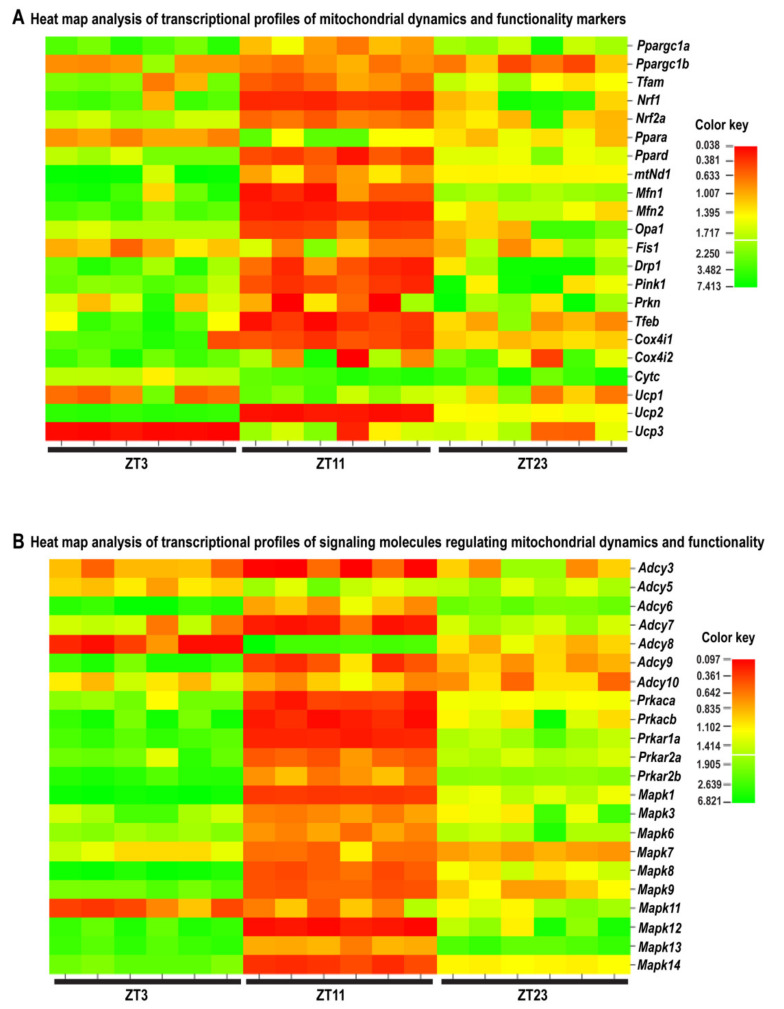
Heat map analysis of the transcriptional profile of mitochondrial dynamics and functionality markers (**A**) as well as signaling molecules regulating mitochondrial dynamics and functionality (**B**) in spermatozoa obtained from adult rats repeatedly stressed at different time points (ZT3, ZT11, and ZT23) during the day. Analysis showing different patterns of transcription at different time points in spermatozoa after repeated immobilization stress. Heat map analysis shows a relative fold of change in gene expression for the aforementioned markers at different time points (ZT3, ZT11, and ZT23), which are presented in colors from red to green, indicating low to high expression.

**Figure 11 cells-11-00993-f011:**
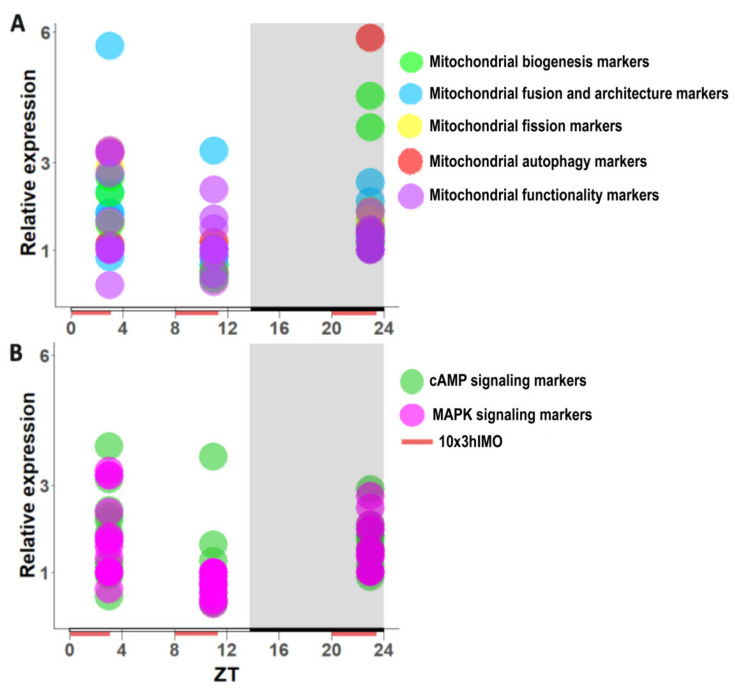
Pattern of transcripts from spermatozoa obtained from repeatedly stressed rats at different ZT time points (ZT3, ZT11, and ZT23) during the day (24 h). Data show transcriptional pattern of genes encoding the proteins are important for mitochondrial dynamics/functionality (**A**) as well as cAMP and MAPK signaling pathways (**B**). Points represent a deviation in the transcription of a particular gene at different ZT time points.

**Figure 12 cells-11-00993-f012:**
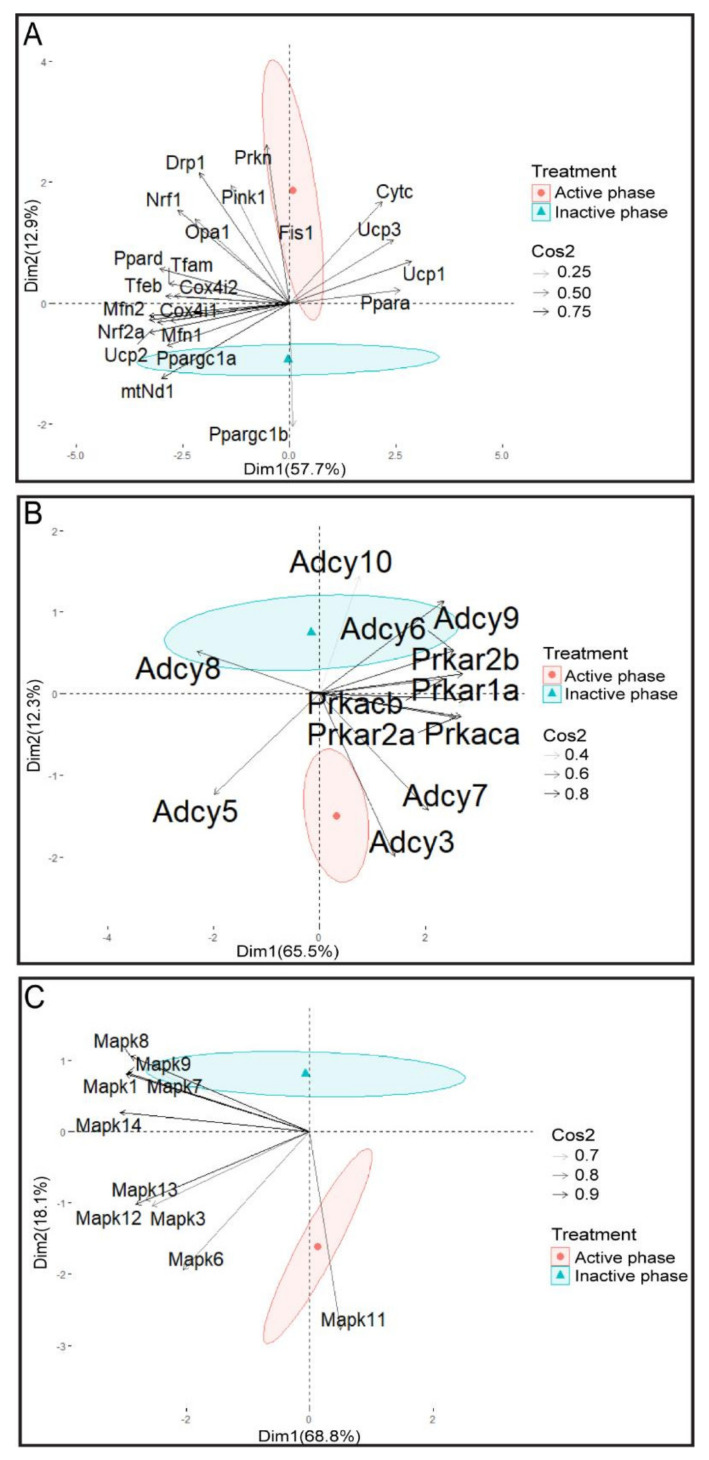
PCA of mitochondrial dynamics (**A**), cAMP signaling pathway (**B**), MAPK signaling pathway, (**C**) and gene expression on active/inactive phase; Dim1 and Dim2 represent the first two PCs and % of the retained variation. Cos2 estimates the qualitative representation of variables.

**Figure 13 cells-11-00993-f013:**
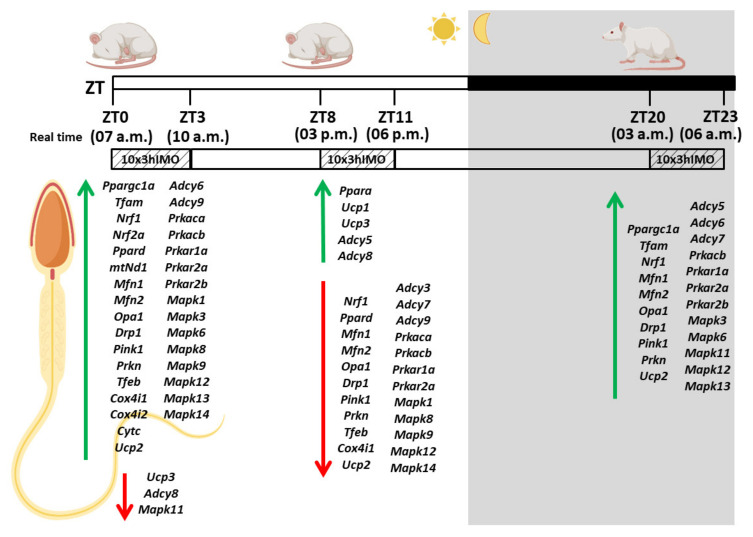
The transcriptional profile of mitochondrial dynamics and functionality markers as well as signaling molecules regulating mitochondrial dynamics and functionality in spermatozoa obtained from adult rats repeatedly stressed at different time points (ZT3, ZT11, and ZT23) during the day. Up arrow symbol represents increased transcription, while down arrow symbol represents decreased transcription.

**Table 1 cells-11-00993-t001:** Transcriptional profiles of mitochondrial dynamics and functionality markers and signaling molecules regulating mitochondrial dynamics and functionality as well as spermatozoa number and functionality in spermatozoa of repeatedly stressed adult rats.

	Time Points					
	ZT3		ZT11		ZT23	
Group Transcript	Control	1×3hIMO	Control	1×3hIMO	Control	1×3hIMO
** *Ppargc1a* **	**1.0** ± 0.07	**3.1** * ± 0.51↑	**1.0** ± 0.06	**1.0** ± 0.17	**1.0** ± 0.07	**2.6** * ± 0.72↑
** *Tfam* **	**1.0** ± 0.07	**1.7** * ± 0.33↑	**1.0** ± 0.05	**0.7** ± 0.09	**1.0** ± 0.07	**1.6** * ± 0.10↑
** *Nrf1* **	**1.0** ± 0.09	**2.7** * ± 0.54↑	**1.0** ± 0.07	**0.3** * ± 0.02↓	**1.0** ± 0.07	**3.1** * ± 0.80↑
** *Nrf2a* **	**1.0** ± 0.06	**1.8** * ± 0.06↑	**1.0** ± 0.05	**0.6** ± 0.06	**1.0** ± 0.06	**1.9** * ± 0.68↑
** *Ppara* **	**1.0** ± 0.07	**0.9** ± 0.04	**1.0** ± 0.05	**2.4** * ± 0.46↑	**1.0** ± 0.07	**1.3** ± 0.11
** *Ppard* **	**1.0** ± 0.06	**2.0** * ± 0.13↑	**1.0** ± 0.06	**0.4** * ± 0.05↓	**1.0** ± 0.07	**1.7** * ± 0.17↑
** *mtNd1* **	**1.0** ± 0.07	**5.7** * ± 1.36↑	**1.0** ± 0.07	**0.9** ± 0.15	**1.0** ± 0.08	**1.4** ± 0.01
** *Mfn1* **	**1.0** ± 0.08	**3.7** * ± 0.55↑	**1.0** ± 0.05	**0.4** * ± 0.12↓	**1.0** ± 0.07	**1.9** * ± 0.04↑
** *Mfn2* **	**1.0** ± 0.06	**3.0** * ± 0.24↑	**1.0** ± 0.05	**0.3** * ± 0.01↓	**1.0** ± 0.07	**1.6** * ± 0.11↑
** *Opa1* **	**1.0** ± 0.09	**1.7** * ± 0.04↑	**1.0** ± 0.06	**0.5** * ± 0.09↓	**1.0** ± 0.09	**2.1** * ± 0.38↑
** *Drp1* **	**1.0** ± 0.04	**3.0** * ± 0.39↑	**1.0** ± 0.04	**0.5** * ± 0.10↓	**1.0** ± 0.07	**4.2** * ± 1.05↑
** *Pink1* **	**1.0** ± 0.06	**2.1** * ± 0.15↑	**1.0** ± 0.05	**0.4** * ± 0.04↓	**1.0** ± 0.07	**3.8** * ± 1.12↑
** *Prkn* **	**1.0** ± 0.05	**2.1** * ± 0.43↑	**1.0** ± 0.07	**0.4** * ± 0.21↓	**1.0** ± 0.04	**2.8** * ± 1.02↑
** *Tfeb* **	**1.0** ± 0.06	**3.1** * ± 0.56↑	**1.0** ± 0.05	**0.3** * ± 0.04↓	**1.0** ± 0.08	**1.2** ± 0.19
** *Cox4i1* **	**1.0** ± 0.07	**3.2** * ± 0.25↑	**1.0** ± 0.06	**0.4** * ± 0.03↓	**1.0** ± 0.07	**1.2** ± 0.05
** *Cox4i2* **	**1.0** ± 0.06	**3.3** * ± 0.58↑	**1.0** ± 0.05	**1.7** * ± 0.91↑	**1.0** ± 0.01	**1.9** * ± 0.61↑
** *Cytc* **	**1.0** ± 0.09	**1.7** * ± 0.09↑	**1.0** ± 0.06	**0.9** ± 0.02	**1.0** ± 0.07	**1.3** ± 0.11
** *Ucp1* **	**1.0** ± 0.06	**1.1** ± 0.42	**1.0** ± 0.07	**2.4** * ± 0.35↑	**1.0** ± 0.07	**1.4** ± 0.29
** *Ucp2* **	**1.0** ± 0.06	**3.8** * ± 0.13↑	**1.0** ± 0.05	**0.3** * ± 0.02↓	**1.0** ± 0.05	**1.5** * ± 0.01↑
** *Ucp3* **	**1.0** ± 0.04	**0.2** * ± 0.02↓	**1.0** ± 0.03	**1.5** * ± 0.32↑	**1.0** ± 0.07	**1.2** ± 0.27
** *Adcy3* **	**1.0** ± 0.05	**0.7** ± 0.12	**1.0** ± 0.06	**0.3** * ± 0.09↓	**1.0** ± 0.07	**1.1** ± 0.30
** *Adcy5* **	**1.0** ± 0.05	**0.9** ± 0.06	**1.0** ± 0.07	**1.6** * ± 0.17↑	**1.0** ± 0.06	**1.5** * ± 0.12↑
** *Adcy6* **	**1.0** ± 0.03	**3.6** * ± 0.34↑	**1.0** ± 0.06	**0.9** ± 0.12	**1.0** ± 0.07	**2.1** * ± 0.08↑
** *Adcy7* **	**1.0** ± 0.02	**1.2** ± 0.20	**1.0** ± 0.05	**0.4** * ± 0.09↓	**1.0** ± 0.07	**1.5** * ± 0.09↑
** *Adcy8* **	**1.0** ± 0.05	**0.3** * ± 0.07↓	**1.0** ± 0.07	**3.6** * ± 1.00↑	**1.0** ± 0.08	**1.0** ± 0.09
** *Adcy9* **	**1.0** ± 0.05	**3.0** * ± 0.32↑	**1.0** ± 0.09	**0.6** ± 0.17	**1.0** ± 0.07	**0.9** ± 0.06
** *Prkaca* **	**1.0** ± 0.04	**1.9** * ± 0.13↑	**1.0** ± 0.07	**0.3** * ± 0.03↓	**1.0** ± 0.07	**1.2** ± 0.01
** *Prkacb* **	**1.0** ± 0.03	**3.1** * ± 0.41↑	**1.0** ± 0.05	**0.3** * ± 0.01↓	**1.0** ± 0.05	**1.8** * ± 0.69↑
** *Prkar1a* **	**1.0** ± 0.05	**2.6** * ± 0.17↑	**1.0** ± 0.06	**0.3** * ± 0.01↓	**1.0** ± 0.08	**1.7** * ± 0.22↑
** *Prkar2a* **	**1.0** ± 0.04	**2.1** * ± 0.27↑	**1.0** ± 0.08	**0.6** ± 0.07	**1.0** ± 0.06	**1.4** ± 0.05
** *Prkar2b* **	**1.0** ± 0.06	**3.1** * ± 0.25↑	**1.0** ± 0.07	**0.7** ± 0.06	**1.0** ± 0.07	**1.8** * ± 0.04↑
** *Mapk1* **	**1.0** ± 0.08	**4.0** * ± 0.09↑	**1.0** ± 0.11	**0.4** * ± 0.01↓	**1.0** ± 0.15	**1.3** ± 0.07
** *Mapk3* **	**1.0** ± 0.07	**2.0** * ± 0.32↑	**1.0** ± 0.10	**0.7** ± 0.03	**1.0** ± 0.14	**1.5** * ± 0.37↑
** *Mapk6* **	**1.0** ± 0.06	**1.8** * ± 0.07↑	**1.0** ± 0.09	**0.7** ± 0.06	**1.0** ± 0.15	**2.0** * ± 0.51↑
** *Mapk8* **	**1.0** ± 0.07	**3.8** * ± 0.19↑	**1.0** ± 0.11	**0.5** * ± 0.04↓	**1.0** ± 0.15	**1.2** ± 0.11
** *Mapk9* **	**1.0** ± 0.04	**2.1** * ± 0.09↑	**1.0** ± 0.10	**0.5** * ± 0.04↓	**1.0** ± 0.14	**1.0** ± 0.11
** *Mapk11* **	**1.0** ± 0.08	**0.6** * ± 0.11↓	**1.0** ± 0.08	**0.9** ± 0.22	**1.0** ± 0.15	**1.5** * ± 0.16↑
** *Mapk12* **	**1.0** ± 0.07	**2.8** * ± 0.39↑	**1.0** ± 0.12	**0.3** * ± 0.01↓	**1.0** ± 0.12	**2.0** * ± 0.54↑
** *Mapk13* **	**1.0** ± 0.05	**2.7** * ± 0.18↑	**1.0** ± 0.11	**0.8** ± 0.04	**1.0** ± 0.15	**2.5** * ± 0.14↑
** *Mapk14* **	**1.0** ± 0.08	**2.2** * ± 0.06↑	**1.0** ± 0.08	**0.3** * ± 0.03↓	**1.0** ± 0.10	**1.1** ± 0.02

Data are presented as means ± SEM values of two independent experiments. Statistical significance at level *p* < 0.05: * vs. control group of each time point. Up arrow symbol represents increased transcription, while down arrow symbol represents decreased transcription.

## Data Availability

All relevant data and samples are available from the corresponding author on request. Further information and requests for resources and reagents should be directed to and will be fulfilled by the Lead Contact, Silvana Andric (silvana.andric@dbe.uns.ac.rs).

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
