# Peer review of "Spermatozoal Mitochondrial Dynamics Markers and Other Functionality-Related Signaling Molecules Exert Circadian-like Response to Repeated Stress of Whole Organism"

_cells, 2022, doi:10.3390/cells11060993_

Round 1

Reviewer 1 Report

This is an interesting study providing insights into the psychophysical stress factors that cause different changes of mitochondrial dynamics and functionality markers and related signaling molecules, depending on the circadian time of the stress exposure. Overall this manuscript is nicely organized and includes a sufficient amount of details in each section that could make it reproducible by other laboratories if needed. The quality of the presented work could be improved by changing the style of some of the figures (especially those that include bar graphs). More specific suggestions for improvements are:
  1. Provide a more precise description for the timeline of the experiment - the text does not include a clear description of what ZT means or stands for (=zero time point?). How do different ZTs correlate with the regular 24hr daily cycle? For example, is ZT11 is it equivalent to 11 am or how to interpret the correlation between each time-point and daily time hour?
  2. Change the color scheme of the bar graph illustrations - the use of red and green is not adding any additional values to the data presented. Probably use of grayscale or some other patterns (white filled bars for control vs. gray diagonal cross line for experimental groups could make it better and remove ambiguity for people with color blindness). 
  3. Some of the figures include letters for the panels (Fig 2, for example) but most others (Fig 3, Fig 4, etc.) are not containing letters for the individual panels. It will improve the quality of the work to add letters and reflect it in the text of the figure legends and main text where if you want to point to specific data set you could address it as "... in Figure 3B for example NRF1 data suggests..." and make the link betting written result description and graphical data set more clear and less ambiguous. 
  4. Some (or all) of the figure text probably should include n= for size groups not only listed as numbers at the top of the bar graphs. Also, it is not clear from the figure text what are the units are at the top of the scanning densitometry bar graphs (for example, Fig 3 data for PGC1 and NRF2 in the center) is that fold change 1 = 100% and 0.9 = 90% (please clarify in the future text)?
  5. In Fig 4 all panels are organized vertically (probably not a good use of space). Please consider horizontal orientation.
  6. Consider including images for GAPDH bands when presenting data normalized to GAPDH.
  7. In table 1 - green and red arrows overlap with numbers. Probably need to reformat and add more spacing.
  8. Consider adding a little discussion on light as a factor in disrupting the circadian rhythm. The methods suggest use of Red light for the time of the stress-inducing procedure. Would you expect more stress if regular or blue light is used for the times/procedures used for disrupting the circadian rhythm?

Author Response

RESPONSES TO REVIEWER#1

Dear Reviewer#1,

We are very grateful to you for giving us the opportunity to revise the manuscript cells-1608465 "Spermatozoal mitochondrial dynamics markers and other functionality-related signaling molecules exert circadian-like response to repeated stress of whole organism” by Starovlah et al.  Our appreciation is much more than we can express even in our mother-tongue-language. We appreciate your great effort, time, patience, and help very much. We accepted all suggestions and revised our manuscript accordingly. Also, we responded point-by-point to all comments. Please see below the main changes in the manuscript (they can be tracked in red in the file named cells-1608465 - MS with Figures - tracked changes) as well as a detailed explanations in point-by-point response to your comments.

(1) The Introduction is improved by adding more details about the aim of the study to explain ZT-points.

(2) The presentation of the results is improved by adding more results related to the protein expression; changing the color scheme; improving the schematic representation of the experimental design (Figure 1) to make ZT-points linked to real-time-points and to clearly show the periods when rats are active (night i.e. dark phase); reorganization of Figure 4 and Table 1, as well as new figures in Supp. Results.

(3) The Results are improved by adding the more results related to the protein expression as well as new figures in Supp. Results.

(4) The Discussion is improved by adding some info related to a light as a stressful factor.

(5) The Methods are improved by adding more info related to the time-line and to clarification of ZT-points.

(6) The References are improved by adding the reference related to a light as a stressful factor.

(7) The Figures were improved by adding the new insert of the results of Western blot analyses; by adding the letters to all figures as well as by reorganization of Figure 4 and Figure 1 (the schematic representation of the experimental design in order to make ZT-points linked to real-time-points and to clearly show the periods when rats are active during the dark phase). Also, the bars are shown in grey scale in order to remove ambiguity for people with color blindness (kind suggestion of Reviewer#1).

(8) The Figure Legends were improved to clearly state the meaning of the numbers in brackets as well as units related to the results of Western blot analyses.

(9) The Supp. Results are improved by adding new figures presenting the results of normalization on control at ZT-3 to show how the transcripts are regulated in a circadian-like fashion in normal controls at different ZT-points.

(10) The English language, grammar and style are improved with help of Mr. Strahinja Keselj (please see acknowledgment) who provides professional peer review support to scholarly publishers, connecting journal editorial offices, authors and reviewers working in R&D-PLANet Systems Group (https://www.planetsg.com/family). He helped also for our manuscript previously published.

We believe that responses below have adequately addressed the questions. All items addressed by Reviewers were accepted and responded specifically. We believe that our results have a significant impact on human/translational physiology, and we anticipate our result to be a starting point for more investigations. Also, we believe our work will be of significant interest to investigators working in the subjects of the basic biology of reproduction and infertility, sex development disorders, and that will provide novel perspectives and approaches, as well as will stimulate the interest of the broad readership of the journal and will be of considerable interest for the diverse readership of the Cells in general.

Please find enclosed Supplementary Material containing the Supp. Results and  Supp. Methods as well as Supplementary Results for review purpose only. 

(1) All authors have made a significant contribution to the paper, have read, and approved the final version of the manuscript for publication and took responsibility for the content and completeness of the manuscript and understood that if the paper, or part of the paper, is found to be faulty or fraudulent, that he/she shares responsibility with his/her coauthors.  

(2) All authors concur with the submission and that the material submitted for publication is an original work, has not been previously published whole or in part, and is not under consideration for publication elsewhere. 

(3) All authors confirm that the work will not be submitted to another journal while under consideration by the Cells.

(4) All authors confirm that this work is not duplicative, neither in the experiment conducted nor the text submitted.

(5) All authors declare that they have no competing financial or other interests and have nothing to disclose.

Please see below point-by-point responses to your comments.

Thank you very much in advance for your time and patience to read the responses as well as the revised manuscript.

With very best regards, for the authors, 

Silvana Andric, corresponding author

SPECIFIC RESPONSES TO REVIEWER#1

Reviewer#1: This is an interesting study providing insights into the psychophysical stress factors that cause different changes of mitochondrial dynamics and functionality markers and related signaling molecules, depending on the circadian time of the stress exposure. Overall, this manuscript is nicely organized and includes a sufficient amount of details in each section that could make it reproducible by other laboratories if needed. The quality of the presented work could be improved by changing the style of some of the figures (especially those that include bar graphs).

RESPONSE: We are really very grateful, and we appreciate your time, careful reading, patience, kindness, and helpful suggestions very much. Our appreciation is much more than we can express even in our mother-tongue-languageJ. We accepted all your suggestions and specifically responded point-by-point to all comments (please see below specific responses for the comments). Also, according to your suggestions listed in the reviewer questionnaire, the presentation of the results and the methods description are improved. In addition, the English grammar, language and style are improved with help of Mr. Strahinja Keselj (please see acknowledgment) who provides professional peer review support to scholarly publishers, connecting journal editorial offices, authors and reviewers working in R&D-PLANet Systems Group (https://www.planetsg.com/family). He helped also for our manuscript previously published (please see the new manuscript as well as the version of the manuscript with “track changes”). Thank you very much in advance for your time and patience to read the responses as well as the revised manuscript.

Reviewer#1 – Comment (1): Provide a more precise description for the timeline of the experiment - the text does not include a clear description of what ZT means or stands for (=zero time point?). How do different ZTs correlate with the regular 24hr daily cycle? For example, is ZT11 is it equivalent to 11 am or how to interpret the correlation between each time-point and daily time hour?

RESPONSE 1: Thank you very much for the helpful comment. The methods description of revised manuscript provides a more precise description for the timeline of the experiment. In addition, as it is stated above, the schematic representation of the experimental design (Figure 1) is improved to link ZT-points to real-time-points and to clearly show the periods when rats are active (night i.e. dark phase).

Reviewer#1 – Comment (2): Change the color scheme of the bar graph illustrations - the use of red and green is not adding any additional values to the data presented. Probably use of grayscale or some other patterns (white filled bars for control vs. gray diagonal cross line for experimental groups could make it better and remove ambiguity for people with color blindness).

RESPONSE 2: Thank you very much for the helpful suggestion. We apologize and we are very sorry that we did not think about issue you kindly mentioned. Namely, we wanted to keep the same colors we used in previously publishes articles. However, certainly we agree with you and we are shame that we did not thing about that. The color scheme of the bar graph illustrations was changed on grayscale and some patterns. As you kindly suggested, white filled bars for control vs. gray diagonal cross line for experimental groups.

Reviewer#1 – Comment (3): Some of the figures include letters for the panels (Fig 2, for example) but most others (Fig 3, Fig 4, etc.) are not containing letters for the individual panels. It will improve the quality of the work to add letters and reflect it in the text of the figure legends and main text where if you want to point to specific data set you could address it as "... in Figure 3B for example NRF1 data suggests..." and make the link betting written result description and graphical data set more clear and less ambiguous.

RESPONSE 3: Thank you very much for the helpful suggestion. The letters are included in the panels and mentioned in the text of the figure legends as well as main text. 

Reviewer#1 – Comment (4): Some (or all) of the figure text probably should include n= for size groups not only listed as numbers at the top of the bar graphs. Also, it is not clear from the figure text what are the units are at the top of the scanning densitometry bar graphs (for example, Fig 3 data for PGC1 and NRF2 in the center) is that fold change 1 = 100% and 0.9 = 90% (please clarify in the future text).

RESPONSE 4: Thank you very much for the helpful suggestion. We apologize for not being clear. The figure legends were improved and include better explanation for the numbers at the top of the bar graphs as well units for the scanning densitometry bar graphs.

Reviewer#1 – Comment (5): In Fig 4 all panels are organized vertically (probably not a good use of space). Please consider horizontal orientation.

RESPONSE 5: Thank you very much for the helpful suggestion. We agree and we apologize for the mistake. The graphs on Figure 4 are reorganized and horizontally oriented.

Reviewer#1 – Comment (6): Consider including images for GAPDH bands when presenting data normalized to GAPDH.

RESPONSE 6: Thank you very much for the helpful suggestion. Certainly, we agree that it GAPDH bands are required. We apologize and we are very sorry for not presenting the GAPDH bands images for initial submission (we rushed to submit before deadline and honestly, we were not aware that we forgot to include GAPDH bands). The images for GAPDH bands are included in the figures of the revised manuscript.

Reviewer#1 – Comment (7): In table 1 - green and red arrows overlap with numbers. Probably need to reformat and add more spacing.

RESPONSE 7: Thank you very much for useful suggestion. We apologize and we are sorry for overlap, but it was not visible in in the versions uploaded and during check-up. We also saw the overlapping when downloaded version for revision (suggestion of editorial office). The more spacing was added in the revised Table 1.

Reviewer#1 – Comment (8): Consider adding a little discussion on light as a factor in disrupting the circadian rhythm. The methods suggest use of Red light for the time of the stress-inducing procedure. Would you expect more stress if regular or blue light is used for the times/procedures used for disrupting the circadian rhythm?

RESPONSE 8: Thank you very much for the suggestion. The revised version of the manuscript includes a little discussion related to light as a stressful factor. Namely, it is well known that light is disrupting the circadian rhythm. The  world-famous laboratories working in in the field of chronobiology suggest use of red light during active/dark phase in experiments with rodents. We agree with you that will be more stressful if regular or blue light is used during the active/dark phase. That is the reason for using red light.

Reviewer 2 Report

The article entitled “Spermatozoal Mitochondrial Dynamics Markers and other Functionality-Related Signaling Molecules Exert Circadian-like Response to Repeated Stress of Whole Organism” is a very interesting manuscript. The study is trying to analyze the spermatozoal mitochondrial role on the fertility problems as a result of stress. On the other hand, the manuscript shows the circadian variation of the stress effect.

The introduction is an excellent review of the “State of the Art”. Materials and methods are accurately described. Results are easy to understand. And the discussion seems very interesting.

Specific Comments

Materials and Methods

Line 181. Were the samples directly stored at -80ºC?

Line 208. In the line 181 you said that the samples were stored at -80ºC and here at -70ºC?

Line 236. The same as in Line 208.

Results

Line 259 to 266 and Fig 1 are Materials and Methods.

Line 278. The expression “Beside well known” could be discussion but not results.

Line 321. “Very well known”, the same.

Line 324. “Interestingly”, the same.

Lines 456 to 459. It’s also discussion.

Discussion

Line 657. “During the day”.

Author Response

RESPONSES TO REVIEWER#2

Dear Reviewer#2,

We are very grateful to you for giving us the opportunity to revise the manuscript cells-1608465 "Spermatozoal mitochondrial dynamics markers and other functionality-related signaling molecules exert circadian-like response to repeated stress of whole organism” by Starovlah et al.  Our appreciation is much more than I can express even in our mother-tongue-language. We appreciate your great effort, time, patience, and help very much. We accepted all suggestions and revised our manuscript accordingly. Also, we responded point-by-point to all comments. Please see below the main changes in the manuscript (they can be tracked in red in the file named cells-1608465 - MS with Figures - tracked changes) as well as a detailed explanations in point-by-point response to your comments.

(1) The Introduction is improved by adding more details about the aim of the study to explain ZT-points.

(2) The presentation of the results is improved by adding more results related to the protein expression; changing the color scheme; improving the schematic representation of the experimental design (Figure 1) to make ZT-points linked to real-time-points and to clearly show the periods when rats are active (night i.e. dark phase); reorganization of Figure 4 and Table 1, as well as new figures in Supp. Results.

(3) The Results are improved by adding the more results related to the protein expression as well as new figures in Supp. Results.

(4) The Discussion is improved by adding some info related to a light as a stressful factor.

(5) The Methods are improved by correcting the mistakes and adding more info related to the time-line and to clarification of ZT-points.

(6) The References are improved by correcting the mistakes and adding the reference related to a light as a stressful factor.

(7) The Figures were improved by adding the new insert of the results of Western blot analyses; by adding the letters to all figures as well as by reorganization of Figure 4 and Figure 1 (the schematic representation of the experimental design in order to make ZT-points linked to real-time-points and to clearly show the periods when rats are active during the dark phase). Also, the bars are shown in grey scale in order to remove ambiguity for people with color blindness (suggestion of Reviewer#1).

(8) The Figure Legends were improved to clearly state the meaning of the numbers in brackets as well as units related to the results of Western blot analyses.

(9) The Supp. Results are improved by adding new figures presenting the results of normalization on control at ZT-3 to show how the transcripts are regulated in a circadian-like fashion in normal controls at different ZT-points.

(10) The English language, grammar and style are improved with help of Mr. Strahinja Keselj (please see acknowledgment) who provides professional peer review support to scholarly publishers, connecting journal editorial offices, authors and reviewers working in R&D-PLANet Systems Group (https://www.planetsg.com/family). He helped also for our manuscript previously published.

We believe that responses below have adequately addressed the questions. All items addressed by Reviewers were accepted and responded specifically. We believe that our results have a significant impact on human/translational physiology, and we anticipate our result to be a starting point for more investigations. Also, we believe our work will be of significant interest to investigators working in the subjects of the basic biology of reproduction and infertility, sex development disorders, and that will provide novel perspectives and approaches, as well as will stimulate the interest of the broad readership of the journal and will be of considerable interest for the diverse readership of the Cells in general.

Please find enclosed Supplementary Material containing the Supp. Results and  Supp. Methods as well as Supplementary Results for review purpose only. 

(1) All authors have made a significant contribution to the paper, have read, and approved the final version of the manuscript for publication and took responsibility for the content and completeness of the manuscript and understood that if the paper, or part of the paper, is found to be faulty or fraudulent, that he/she shares responsibility with his/her coauthors.  

(2) All authors concur with the submission and that the material submitted for publication is an original work, has not been previously published whole or in part, and is not under consideration for publication elsewhere. 

(3) All authors confirm that the work will not be submitted to another journal while under consideration by the Cells.

(4) All authors confirm that this work is not duplicative, neither in the experiment conducted nor the text submitted.

(5) All authors declare that they have no competing financial or other interests and have nothing to disclose.

Please see below point-by-point responses to your comments.

Thank you very much in advance for your time and patience to read the responses as well as the revised manuscript.

With very best regards, for the authors, 

Silvana Andric, corresponding author

SPECIFIC RESPONSES TO REVIEWER#2

 Reviewer#2: The article entitled “Spermatozoal Mitochondrial Dynamics Markers and other Functionality-Related Signaling Molecules Exert Circadian-like Response to Repeated Stress of Whole Organism” is a very interesting manuscript. The study is trying to analyze the spermatozoal mitochondrial role on the fertility problems as a result of stress. On the other hand, the manuscript shows the circadian variation of the stress effect. The introduction is an excellent review of the “State of the Art”. Materials and methods are accurately described. Results are easy to understand. And the discussion seems very interesting.

RESPONSE: We are really very grateful, and we appreciate your time, careful reading, patience, kindness, and helpful suggestions very much. Our appreciation is much more than we can express even in our mother-tongue-languageJ. We accepted all your suggestions and specifically responded point-by-point to all comments (please see below specific responses for the comments). Thank you very much in advance for your time and patience to read the responses as well as the revised manuscript.

Reviewer#2 – Comment (1): Line 181. Were the samples directly stored at -80ºC? Line 208. In the line 181 you said that the samples were stored at -80ºC and here at -70ºC?

Line 236. The same as in Line 208

RESPONSE 1: Thank you very much for the helpful comment. We apologize and we are very sorry for the mistake. The samples were stored at -80ºC. The mistake is corrected.

Reviewer#2 – Comment (2): Line 259 to 266 and Fig 1 are Materials and Methods.

RESPONSE 2: RESPONSE 2: Thank you very much for the helpful comment and we apologize for the mistake. Figure 1 was “moved” in Materia and Methods. The related-text describing shortly the experimental design is for the better understanding of the results since some readers do not read Material and Methods. We believe that it will facilitate the understanding of the results.

Reviewer#2 – Comment (3):

Line 278. The expression “Beside well known” could be discussion but not results.

Line 321. “Very well known”, the same.

Line 324. “Interestingly”, the same.

Lines 456 to 459. It’s also discussion.

RESPONSE 3a: Thank you very much for the helpful comment. We apologize and we are sorry for the making confusion. They are corrected in the revised manuscript.

Lines 456 to 459. It’s also discussion.

RESPONSE 3b: We believe that this will facilitate understanding of the purpose of performing the analyses.

Reviewer#2 – Comment (4): Line 657. “During the day”.

RESPONSE 4: Thank you very much for the helpful comment and we are sorry about the mistake. It is corrected in the revised manuscript.

Reviewer 3 Report

The manuscript by Starovlah et al investigates transcriptional changes occurring in rat spermatozoa under stress. The authors focus on transcripts encoding for mitochondria related proteins at different time points, in order to uncover a dysregulation of circadian rhythm. They detected a major impairment in the expression profiles of vast majority of analyzed genes, thus suggesting an impact of stress condition on circadian-like-adaptive-mechanisms. Overall, the manuscript is reasonably presented, and the goal is clearly explained. The experimental approach is somewhat limited, given that an unbiased transcriptomic analysis through RNAseq would have been significantly more informative (even probably requiring lower experimental effort). In any case, these data are new and worth to be reported. I have a couple of suggestions to improve the manuscript:

  • All data are normalized against control condition. Although I understand this choice, this kind of analysis does not show whether and how these transcripts are regulated in a circadian-like fashion in normal controls.
  • I think there are some problems in densitometric analyses of Western blots. The provided images do not reflect the quantification provided in corresponding bar graph. On the one hand, this is likely due to the fact that data in bar graphs are normalized on GADPH, that is not shown (it must be included). On the other hand, this probably reveals major differences in GADPH levels, maybe indicating sub-optimal quantification of samples.

Author Response

RESPONSES TO REVIEWER#3

Dear Reviewer#3,

We are very grateful to you for giving us the opportunity to revise the manuscript cells-1608465 "Spermatozoal mitochondrial dynamics markers and other functionality-related signaling molecules exert circadian-like response to repeated stress of whole organism” by Starovlah et al.  Our appreciation is much more than I can express even in our mother-tongue-language. We appreciate your great effort, time, patience, and help very much. We accepted all suggestions and revised our manuscript accordingly. Also, we responded point-by-point to all comments. Please see below the main changes in the manuscript (they can be tracked in red in the file named cells-1608465 - MS with Figures - tracked changes) as well as a detailed explanations in point-by-point response to your comments.

(1) The Introduction is improved by adding more details about the aim of the study to explain ZT-points.

(2) The presentation of the results is improved by adding more results related to the protein expression; changing the color scheme; improving the schematic representation of the experimental design (Figure 1) to make ZT-points linked to real-time-points and to clearly show the periods when rats are active (night i.e. dark phase); reorganization of Figure 4 and Table 1, as well as new figures in Supp. Results.

(3) The Results are improved by adding the more results related to the protein expression as well as new figures in Supp. Results.

(4) The Discussion is improved by adding some info related to a light as a stressful factor.

(5) The Methods are improved by adding more info related to the time-line and to clarification of ZT-points.

(6) The References are improved by adding the reference related to a light as a stressful factor.

(7) The Figures were improved by adding the new insert of the results of Western blot analyses; by adding the letters to all figures as well as by reorganization of Figure 4 and Figure 1 (the schematic representation of the experimental design in order to make ZT-points linked to real-time-points and to clearly show the periods when rats are active during the dark phase). Also, the bars are shown in grey scale in order to remove ambiguity for people with color blindness (suggestion of Reviewer#1).

(8) The Figure Legends were improved to clearly state the meaning of the numbers in brackets as well as units related to the results of Western blot analyses.

(9) The Supp. Results are improved by adding new figures presenting the results of normalization on control at ZT-3 to show how the transcripts are regulated in a circadian-like fashion in normal controls at different ZT-points.

(10) The English language, grammar and style are improved with help of Mr. Strahinja Keselj (please see acknowledgment) who provides professional peer review support to scholarly publishers, connecting journal editorial offices, authors and reviewers working in R&D-PLANet Systems Group (https://www.planetsg.com/family). He helped also for our manuscript previously published.

We believe that responses below have adequately addressed the questions. All items addressed by Reviewers were accepted and responded specifically. We believe that our results have a significant impact on human/translational physiology, and we anticipate our result to be a starting point for more investigations. Also, we believe our work will be of significant interest to investigators working in the subjects of the basic biology of reproduction and infertility, sex development disorders, and that will provide novel perspectives and approaches, as well as will stimulate the interest of the broad readership of the journal and will be of considerable interest for the diverse readership of the Cells in general.

Please find enclosed Supplementary Material containing the Supp. Results and  Supp. Methods as well as Supplementary Results for review purpose only. 

(1) All authors have made a significant contribution to the paper, have read, and approved the final version of the manuscript for publication and took responsibility for the content and completeness of the manuscript and understood that if the paper, or part of the paper, is found to be faulty or fraudulent, that he/she shares responsibility with his/her coauthors.  

(2) All authors concur with the submission and that the material submitted for publication is an original work, has not been previously published whole or in part, and is not under consideration for publication elsewhere. 

(3) All authors confirm that the work will not be submitted to another journal while under consideration by the Cells.

(4) All authors confirm that this work is not duplicative, neither in the experiment conducted nor the text submitted.

(5) All authors declare that they have no competing financial or other interests and have nothing to disclose.

Please see below point-by-point responses to your comments.

Thank you very much in advance for your time and patience to read the responses as well as the revised manuscript.

With very best regards, for the authors, 

Silvana Andric, corresponding author

SPECIFIC RESPONSES TO REVIEWER#3

Reviewer#3: The manuscript by Starovlah et al investigates transcriptional changes occurring in rat spermatozoa under stress. The authors focus on transcripts encoding for mitochondria related proteins at different time points, in order to uncover a dysregulation of circadian rhythm. They detected a major impairment in the expression profiles of vast majority of analyzed genes, thus suggesting an impact of stress condition on circadian-like-adaptive-mechanisms. Overall, the manuscript is reasonably presented, and the goal is clearly explained. The experimental approach is somewhat limited, given that an unbiased transcriptomic analysis through RNAseq would have been significantly more informative (even probably requiring lower experimental effort). In any case, these data are new and worth to be reported. I have a couple of suggestions to improve the manuscript.

RESPONSE: We are really very grateful, and we appreciate your time, careful reading, patience, kindness, and helpful suggestions very much. Our appreciation is much more than we can express even in our mother-tongue-languageJ. We agree with you that experimental approach is somewhat limited and that an unbiased transcriptomic analysis through RNAseq would have been significantly more informative and requiring lower experimental effort, but we did what we can accomplish in our country with extremely limited funds. Certainly, we will try to find colleagues abroad interested in collaboration to run RNAseq. We accepted all other suggestions and specifically responded point-by-point to all comments (please see below specific responses for the comments). Thank you very much in advance for your time and patience to read the responses as well as the revised manuscript.

Reviewer#3 – Comment (1): All data are normalized against control condition. Although I understand this choice, this kind of analysis does not show whether and how these transcripts are regulated in a circadian-like fashion in normal controls.

RESPONSE 1: Thank you very much for the helpful comment. We agree with you that the using corresponding controls as a calibrator does not show whether and how these transcripts are regulated in a circadian-like fashion.  We did that analyses and graphs in parallel with those normalized on corresponding control, but we were afraid that will make confusion and since deadline for special issue was limited, we did not show it. However, in the revised manuscript we show results in supplementary because we are afraid that more figures in the main text will make confusion. The results are commented in sections Results and Discussion of the revised manuscript.

Reviewer#3 – Comment (2): I think there are some problems in densitometric analyses of Western blots. The provided images do not reflect the quantification provided in corresponding bar graph. On the one hand, this is likely due to the fact that data in bar graphs are normalized on GADPH, that is not shown (it must be included). On the other hand, this probably reveals major differences in GADPH levels, maybe indicating sub-optimal quantification of samples.

RESPONSE 2: Certainly, we agree that it images of GAPDH bands are required. We apologize and we are very sorry for not presenting the GAPDH bands images for initial submission (we rushed to submit before deadline and honestly, we were not aware that we forgot to include GAPDH bands). We agree with you with all you mentioned. Searching for the reason it appears that there was problem with spectrophotometer filter and measurement of proteins was not correct. Accordingly, we run during given time 32 new membranes and obtained accurate GAPDH and some new proteins. The images for GAPDH bands and other proteins of interest are included in the figures of the revised manuscript.

Round 2

Reviewer 3 Report

The authors succesfully addessed all my origina concerns. I think the manuscript is improved and deserves publication